# Analytic conformal bootstrap and Virasoro primary fields in the Ashkin–Teller model

**Nikita Nemkov**[1] **and Sylvain Ribault**[2]

[1] *Russian Quantum Center, Skolkovo, Moscow 143025, Russia*
[2] *Université Paris-Saclay, CNRS, CEA, Institut de physique théorique*

*E-mail:* `nnemkov@gmail.com, sylvain.ribault@ipht.fr`

ABSTRACT: We revisit the critical two-dimensional Ashkin–Teller model, i.e. the $\mathbb{Z}_2$ orbifold of the compactified free boson CFT at $c = 1$. We solve the model on the plane by computing its three-point structure constants and proving crossing symmetry of four-point correlation functions. We do this not only for affine primary fields, but also for Virasoro primary fields, i.e. higher twist fields and degenerate fields.

This leads us to clarify the analytic properties of Virasoro conformal blocks and fusion kernels at $c = 1$. We show that blocks with a degenerate channel field should be computed by taking limits in the central charge, rather than in the conformal dimension. In particular, Al. Zamolodchikov's simple explicit expression for the blocks that appear in four-twist correlation functions is only valid in the non-degenerate case: degenerate blocks, starting with the identity block, are more complicated generalized theta functions.

# 1 Introduction

After Belavin, Polyakov and Zamolodchikov worked out the basics of the conformal bootstrap approach to two-dimensional CFT [1], the critical Ashkin–Teller model was one of the first theories to be solved. The model differs from the compactified free boson by its twist sector, whose correlation functions were exactly computed by Al. Zamolodchikov [2]. One may think that there is little left to say on the subject. This would be quite wrong, as we will now argue.

## Affine symmetry versus Virasoro symmetry

Per the conformal bootstrap's basic doctrine, the model's solution extensively relies on its affine symmetry algebra. In practice, this means that only correlation functions of affine primary fields are explicitly known. Other correlation functions are in principle determined by the symmetry, which makes them accessible to pedestrian calculations on a case-by-case basis. However, the model contains infinitely many Virasoro primary fields that are not affine primary fields, and it would be very interesting to determine their correlation functions in general.

In particular, a special case of the Ashkin–Teller model is supposed to coincide with the 4-state Potts model [3]. The $Q$-state Potts model does not have affine symmetry for generic $Q \in \mathbb{C}$, and solving it means computing correlation functions of Virasoro primary fields. To compare it with the Ashkin–Teller model, we need to understand the latter's Virasoro primary fields.

Even in the free boson CFT, this issue has not been solved. There is an infinite series of Virasoro primary fields, called degenerate fields, which are affine descendants of the identity field. Their correlation functions are not known. Nevertheless, boundary conditions that preserve Virasoro symmetry only (and break affine symmetry) have been determined by Janik [4].

In the Ashkin–Teller model, in addition to degenerate fields, there is another infinite series of Virasoro primary fields, called higher twist fields. Some correlation functions that involve the first few higher twist fields were computed by Apikyan and Al. Zamolodchikov [5], using the model's affine symmetry. Their method is based on affine symmetry, and cannot work for arbitrary higher twist fields.

We will develop a method that does not rely explicitly on the affine symmetry. Technically, we will solve crossing symmetry equations, after formulating them in terms of fusion kernels rather than conformal blocks. The relevant fusion kernels can be computed explicitly, which will allow us to determine the structure constants of arbitrary Virasoro primary fields.

## Subtleties of Virasoro conformal blocks and fusion kernels

In order to write crossing symmetry equations, we need to know the relevant Virasoro conformal blocks or fusion kernels. The basic techniques for computing the blocks date back to the 1980s: BPZ differential equations [1], and Al. Zamolodchikov's recursive representation [6]. However, these techniques work better at generic values of the central charge. For $c = 1$ and other rational values, there appear Virasoro representations with intricate structures, i.e. infinite ladders of singular vectors. This leads to conformal blocks having multiple poles as functions of the channel dimension, rather than simple poles. Worse, the recursive representation breaks down.

This would not be a big problem if our CFT only involved Verma modules of the Virasoro algebra. For example, in the case of Liouville theory at $c = 1$, conformal blocks can be computed by slightly perturbing the central charge [7], or by using $c = 1$ expressions deduced from the Painlevé VI equation [8]. However, the Ashkin–Teller model and the free boson CFT also include degenerate representations, whose dimensions fall on the poles of conformal blocks. Of course, conformal blocks must actually be finite, even when they involve degenerate representations. So the residues of the poles must vanish, which indeed occurs in particular correlation functions. The problem is that when computing degenerate blocks as limits from the generic case, the result depends on how we take limits of the various parameters.

We will demonstrate that degenerate blocks can be computed as limits of generic blocks

if $c$ is generic, but not if $c$ is rational. Therefore, to compute our degenerate conformal blocks at $c = 1$, we will have to first find appropriate continuations to generic $c$, and then take the limit $c \to 1$. Degenerate fusion kernels can also be computed using the same procedure. This will allow us to write and solve enough crossing symmetry equations for determining all structure constants of the Ashkin–Teller model. In a number of cases, these structure constants can alternatively be computed from the affine symmetry, which provides independent checks of our ideas on conformal blocks.

**Main results**

As summarized in Table (4.9). the Virasoro primary fields of the Ashkin–Teller model come in three sectors: vertex $V$, identity $I$ (i.e. degenerate fields), and twist $T$. We have determined all fusion rules and two- and three-point structure constants of these fields, in particular:

- The fusion rules of degenerate Virasoro primary fields $I \times I$ (3.17), $I \times T$ (4.15), and $T \times T$ (4.16).

- The chiral three-point structure constants of the types $\langle IVV \rangle$ (3.42), $\langle III \rangle$ (3.49), $\langle VTT \rangle$ (4.47), and $\langle ITT \rangle$ (4.56).

Moreover, the behaviour of some four-point functions suggests that higher twist fields could have a geometrical interpretation, see Eq. (4.65). On degenerate conformal blocks, our results include:

- Conjecture 1 and Conjecture 2 for computing degenerate conformal blocks, at generic and rational central charges respectively.

- Recursive representations (5.24) and explicit expressions as generalized theta functions (5.25) for a class of degenerate blocks at $c = 1$.

Let us also mention some technical results on fusion kernels:

- The fusion kernels for four-point functions with one level 3 degenerate field at generic central charge (5.38)-(5.41).

- The discrete fusion transformation (B.23) for conformal blocks that appear in four-point functions of the type $\langle VVTT \rangle$.

# 2    Basic structures of two-dimensional CFT

In the conformal bootstrap approach, a two-dimensional CFT is a set of correlation functions that obey certain relations. We will review correlation functions and their properties, while paying particular attention to the nontrivial signs that appear when fields have nonzero conformal spins. For a more detailed review, see [9].

## 2.1    Fields and correlation functions

Let $V_{\Delta, \bar{\Delta}}(z)$ be a primary field on the Riemann sphere $z \in (\mathbb{C} \cup \infty)$, with the conformal dimensions $\Delta$ and $\bar{\Delta}$ for the left and right Virasoro algebras. We will assume that our fields have integer conformal spins,

$$S(V_{\Delta, \bar{\Delta}}) = \Delta - \bar{\Delta} \in \mathbb{Z} . \tag{2.1}$$

Correlation functions $\left\langle \prod_{i=1}^n V_i(z_i) \right\rangle = \left\langle \prod_{i=1}^n V_{\Delta_i, \bar{\Delta}_i}(z_i) \right\rangle$ are single-valued functions of $z_i$, and invariant under field permutations,

$$V_1(z_1)V_2(z_2) = V_2(z_2)V_1(z_1) \ . \tag{2.2}$$

Conformal symmetry determines how two- and three-point functions depend on the positions. Let us analyze these correlation functions in some detail. A two-point function can be nonzero only if the two fields have the same left and right conformal dimensions, $\langle V_1 V_2 \rangle \neq 0 \implies (\Delta_1, \bar{\Delta}_1) = (\Delta_2, \bar{\Delta}_2)$. By choosing an appropriate basis $\{V_i\}$ of primary fields, we can further assume

$$\langle V_i V_j \rangle \neq 0 \implies i = j^* \ , \tag{2.3}$$

for some involution $i \to i^*$ such that $(\Delta_i, \bar{\Delta}_i) = (\Delta_{i^*}, \bar{\Delta}_{i^*})$. Often $i^* = i$ but for the free boson it is convenient to choose this involution as the natural $\mathbb{Z}_2$ symmetry of the model. Then the non-vanishing two-point functions are of the type

$$\left\langle V_i(z_1)V_{i^*}(z_2) \right\rangle = \frac{B_i}{\left| z_{12}^{2\Delta_i} \right|^2} \ , \tag{2.4}$$

where the $z_i$-independent factor $B_i = B_{i^*}$ is called the two-point structure constant, and we used the notation

$$\left| z^\Delta \right|^2 \equiv z^\Delta \bar{z}^{\bar{\Delta}} \ . \tag{2.5}$$

Thanks to the spin being integer, this two-point function is invariant under permuting the two fields. It is also single-valued, in particular $z_{12} \to e^{2\pi i} z_{12}$ generates a factor $(-)^{4\Delta_i - 4\bar{\Delta}_i} = 1$. In a unitary CFT, according to the axiom of reflection positivity, we must have $\left\langle V_i(z)V_{i^*}(\bar{z}) \right\rangle \geq 0$, which implies $(-)^{\Delta_i - \bar{\Delta}_i} B_i \geq 0$. Therefore, the two-point structure constant is positive for fields of even spin, and negative for fields of odd spin.

In the case of three-point functions, we have

$$\left\langle V_1(z_1)V_2(z_2)V_3(z_3) \right\rangle = \frac{\delta_{123} C_{123}}{\left| z_{12}^{\Delta_1 + \Delta_2 - \Delta_3} z_{23}^{\Delta_2 + \Delta_3 - \Delta_1} z_{13}^{\Delta_3 + \Delta_1 - \Delta_2} \right|^2} \ , \tag{2.6}$$

where $C_{123}$ is a three-point structure constant, and $\delta_{123} \in \{0, 1\}$ a prefactor that imposes fusion rules if need be. (Distinguishing this prefactor from the structure constant is conceptually clearer, and leads to simpler expressions for the structure constant [10].) Our standard-looking formula for the three-point function hides a subtlety: we have written $z_{13}^{\Delta_3 + \Delta_1 - \Delta_2}$ rather than $z_{31}^{\Delta_3 + \Delta_1 - \Delta_2}$, which changes the overall sign since $\left| z^\Delta \right|^2 = (-)^{\Delta - \bar{\Delta}} \left| (-z)^\Delta \right|^2$. This will lead to the absence of sign factors in the relation (2.10) between OPE coefficients and three-point structure constants.

The three-point function (2.6) is manifestly single-valued, for example it is invariant under $z_{12} \to e^{2\pi i} z_{12}$. Its invariance under permutations is however less obvious, since the $z_i$-dependent factor picks the sign $(-)^{S_1 + S_2 + S_3}$ under odd permutations. For the three-point function to be invariant, we need the three-point structure constant to pick this sign as well,

$$C_{\sigma(1)\sigma(2)\sigma(3)} = |\sigma|^{S_1 + S_2 + S_3} C_{123} \ , \tag{2.7}$$

where $|\sigma|$ is the parity of the permutation $\sigma$.

## 2.2 Conformal bootstrap

**Operator product expansion**

The crucial axiom that leads to nontrivial relations between correlation functions is the existence of an operator product expansion (OPE). In the case of primary fields, the OPE reads

$$V_1(z_1)V_2(z_2) = \sum_{V_i \in \mathcal{S}_{12}} C_{12}^i \left| z_{12}^{\Delta_i - \Delta_1 - \Delta_2} \right|^2 \left( V_i(z_2) + O(z_{12}) \right) , \tag{2.8}$$

where the $z_i$-independent constants $C_{12}^i$ are OPE coefficients, and the set of primary fields $\mathcal{S}_{12}$ is the OPE spectrum. An OPE can be schematically written as a fusion rule for formal fields,

$$V_1 \times V_2 \sim \sum_{i \in \mathcal{S}_{12}} V_i , \tag{2.9}$$

where we omit the positions, OPE coefficients, and subleading terms. Inserting the OPE in a three-point function, we obtain the expression of OPE coefficients in terms of two- and three-point structure constants,

$$C_{12}^3 = \frac{C_{123^*}}{B_3} . \tag{2.10}$$

In the presence of spinful fields, this relation could in principle contain extra sign factors, depending on the precise definitions of the structure constants and OPE coefficients. With our conventions, such sign factors do not appear.

**Decomposing four-point functions into conformal blocks**

We can use OPEs to decompose $n$-point functions into three-point functions. To prove consistency of a CFT on the sphere, it is enough to show that all possible decompositions of 4-point functions agree. We will therefore focus on 4-point functions, starting with the $s$-channel decomposition, which is obtained by inserting the OPE (2.8) of the first two fields,

$$\left\langle \prod_{i=1}^4 V_i(z_i) \right\rangle = \sum_{V_s \in \mathcal{S}_{12} \cap \mathcal{S}_{34}^*} D_{s|1234}^{(s)} \left| \mathcal{F}_{s|1234}^{(s)} \right|^2 , \tag{2.11}$$

where we introduced four-point structure constants

$$D_{s|1234}^{(s)} = \frac{C_{12s^*} C_{s34}}{B_s} . \tag{2.12}$$

and the $s$-channel conformal blocks $\mathcal{F}_{s|1234}^{(s)}$ are implicitly defined by (2.11). In our notations for four-point structure constants and conformal blocks, do not confuse the superscript $(s) \in \{(s), (t), (u)\}$ which only indicates the considered channel, with the subscript $s$ which stands for the channel field with all its properties, including its conformal dimension. Substituting (2.8) into (2.11), we find the asymptotic behaviour of conformal block as $z_1 \to z_2$,

$$\mathcal{F}_{s|1234}^{(s)}(z_1, z_2, z_3, z_4) \underset{z_{12} \to 0}{=} z_{12}^{\Delta_s - \Delta_1 - \Delta_2} \left( z_{23}^{\Delta_4 - \Delta_s - \Delta_3} z_{34}^{\Delta_s - \Delta_3 - \Delta_4} z_{24}^{\Delta_3 - \Delta_4 - \Delta_s} + O(z_{12}) \right) . \tag{2.13}$$

It is often convenient to work with functions of one variable instead of four positions by setting

$$(z_1, z_2, z_3, z_4) = (x, 0, \infty, 1) \ , \tag{2.14}$$

where the field at infinity is defined by $V_{\Delta,\bar{\Delta}}(\infty) = \lim_{z\to\infty} |z^{2\Delta}|^2 V_{\Delta,\bar{\Delta}}(z)$. The conformal blocks then behave as

$$\mathcal{F}^{(s)}_{s|1234}(x) \underset{x\to 0}{=} x^{\Delta_s - \Delta_1 - \Delta_2} \left(1 + O(x)\right) \ . \tag{2.15}$$

If we inserted the OPE $V_2(z_2)V_3(z_3)$ instead of $V_1(z_1)V_2(z_2)$ in the correlation function (2.11), we would obtain the $t$-channel decomposition instead of the $s$-channel decomposition, with the $t$-channel four-point structure constants

$$D^{(t)}_{t|1234} = \frac{C_{23t^*} C_{t41}}{B_t} \ . \tag{2.16}$$

Since the definitions of the $s$ and $t$ channels are related to each other by a permutation of the four fields, we have the relations

$$D^{(t)}_{s|3214} = D^{(s)}_{s|1234} \ , \qquad\qquad \mathcal{F}^{(t)}_{s|3214} = \mathcal{F}^{(s)}_{s|1234} \ . \tag{2.17}$$

**Crossing symmetry**

The equality of the $s$- and $t$-channel decompositions is called crossing symmetry, and reads

$$\sum_{V_s \in \mathcal{S}_{12} \cap \mathcal{S}^*_{34}} D^{(s)}_{s|1234} \left| \mathcal{F}^{(s)}_{s|1234} \right|^2 = \sum_{V_t \in \mathcal{S}_{23} \cap \mathcal{S}^*_{41}} D^{(t)}_{t|1234} \left| \mathcal{F}^{(t)}_{t|1234} \right|^2 \ . \tag{2.18}$$

Given the spectrums $\mathcal{S}_{ij}$, crossing symmetry is a system of non-linear equations for the three-point structure constants.

In the present paper it will be more convenient to use a different form of the crossing symmetry equations. The $s$- and $t$-channel conformal blocks provide two bases of the same space of solutions of conformal Ward identities. The change of bases is a linear relation called the fusion relation,

$$\mathcal{F}^{(s)}_{\Delta_s|\Delta_1,\Delta_2,\Delta_3,\Delta_4} = \sum_{\Delta_t} F_{\Delta_s,\Delta_t} \begin{bmatrix} \Delta_2 & \Delta_3 \\ \Delta_1 & \Delta_4 \end{bmatrix} \mathcal{F}^{(t)}_{\Delta_t|\Delta_1,\Delta_2,\Delta_3,\Delta_4} \ , \tag{2.19}$$

where the $z_i$-independent quantity $F_{\Delta_s,\Delta_t} \begin{bmatrix} \Delta_2 & \Delta_3 \\ \Delta_1 & \Delta_4 \end{bmatrix}$ is called the fusion kernel. Here we label conformal blocks and the fusion kernel by conformal dimensions, in order to emphasize that they only depend on these kinematic quantities. In contrast, fields are not always completely determined by their left and right conformal dimensions. The fusion relation allows to eliminate conformal blocks from the crossing symmetry equation (2.18) in favor of the fusion kernel. Schematically

$$\forall \Delta_t, \bar{\Delta}_t \ , \qquad \sum_{V_s \in \mathcal{S}_{12} \cap \mathcal{S}^*_{34}} D^{(s)}_s F_{\Delta_s,\Delta_t} F_{\bar{\Delta}_s,\bar{\Delta}_t} = \sum_{V_t \in (\mathcal{S}_{23} \cap \mathcal{S}^*_{41})_{\Delta_t,\bar{\Delta}_t}} D^{(t)}_t \ , \tag{2.20}$$

where $(\mathcal{S}_{23} \cap \mathcal{S}^*_{41})_{\Delta_t,\bar{\Delta}_t}$ denotes the subset of fields in the spectrum $\mathcal{S}_{23} \cap \mathcal{S}^*_{41}$ that have left and right dimensions $\Delta_t, \bar{\Delta}_t$. If this subset is made of only one field, then the right-hand side reduces to the four-point structure constant $D^{(t)}_t$ of that field. If this subset is empty, the right-hand side is zero. We insist that Eq. (2.20) holds for any values of $\Delta_t, \bar{\Delta}_t$, even if our model does not contain any field with these dimensions. For example, although Liouville theory is diagonal i.e. all its fields have $\Delta = \bar{\Delta}$, crossing symmetry equations with $\Delta_t \neq \bar{\Delta}_t$ play an important role in analytically solving the theory [9].

## 2.3 Degenerate fields

**Definition and fusion rules**

A primary field that has a vanishing null vector is called a degenerate field. Correlation functions of degenerate fields obey BPZ linear differential equations, and the fusion rules of degenerate fields are particularly simple. Let us write these fusion rules for a central charge $c = 1$. The natural variable for writing fusion rules is not the conformal dimension $\Delta$, but the momentum $p$ such that

$$\Delta = p^2 \ . \tag{2.21}$$

At $c = 1$ degenerate fields have half-integer momentums $k \in \frac{1}{2}\mathbb{Z}$. We will denote $I_k$ a degenerate field with momentum $k$, because in the free boson and orbifold CFTs, degenerate fields are affine descendants of the identity field $I$.

The chiral fusion rule for a degenerate field $I_k$ with another primary field $V_p$ is

$$I_k \times V_p \sim \sum_{i=-k}^{k} V_{p+i} \quad , \quad (k \in \tfrac{1}{2}\mathbb{Z}) \ , \tag{2.22}$$

where the sum runs by increments of 1. For two degenerate fields, we have

$$I_{k_1} \times I_{k_2} \sim \sum_{k=|k_1-k_2|}^{k_1+k_2} I_k \ . \tag{2.23}$$

In particular, the left- and right-degenerate field $I_{0,0}$ enjoys the trivial fusion rule

$$I_{0,0} \times V_{p,\bar{p}} \sim V_{p,\bar{p}} \ . \tag{2.24}$$

We identify $I \equiv I_{0,0}$ with the identity field: a field whose insertion in a correlation function does not change that correlation function.

**Role in solving Liouville theory**

Crossing symmetry equations for four-point functions that involve degenerate fields are enough for determining the three-point structure constants of Liouville theory [11, 7], even though degenerate fields are unphysical, i.e. they do not appear in OPEs. In the case $c = 1$, the Liouville two- and three-point structure constants read

$$B_p^L = 1 \quad , \quad C_{p_1,p_2,p_3}^L = \frac{\prod_{\pm,\pm,\pm} G(1 \pm p_1 \pm p_2 \pm p_3)}{\prod_{i=1}^{3} \prod_{\pm} G(1 \pm 2p_i)} \ . \tag{2.25}$$

Here $p_i = \bar{p}_i$ are momentums of the fields (Liouville theory is diagonal), and $G$ is Barnes' $G$-function, whose properties we will shortly summarize.

Degenerate fields can also be used for constraining or even determining three-point structure constants in non-diagonal CFTs [12, 13]. Under the assumption that two independent degenerate fields exist, it was found that non-diagonal structure constants are related to Liouville theory structure constants by the geometric mean relation

$$C_{V_{p_1,\bar{p}_1} V_{p_2,\bar{p}_2} V_{p_3,\bar{p}_3}} = \pm \sqrt{C_{p_1,p_2,p_3}^L C_{\bar{p}_1,\bar{p}_2,\bar{p}_3}^L} \ . \tag{2.26}$$

Due to the square root, this relation has a sign ambiguity, and does not fully determine the non-diagonal structure constants. Moreover, due to the zeros of $G(x)$ for $x \in -\mathbb{N}$, the Liouville structure constants themselves can be ambiguous for certain values of the momentums, for example when all three momentums are integer. While we will not use the geometric mean relation for solving the Ashkin–Teller model, we will find structure constants that can be written using the same $G$-function.

**Barnes' $G$-function**

At $c = 1$, structure constants and fusion kernels are expressed in terms of Barnes' $G$-function. This holds in the case of Liouville theory, and also of the Ashkin–Teller model, as we will see. So let us review a few properties of this function.

Barnes' $G$-function is an analytic function on the complex plane, with a zero of order $k + 1$ at $z = -k$ for any $k \in \mathbb{N}$. It obeys in particular

$$G(z + 1) = \Gamma(z)G(z) \qquad , \qquad G(1) = 1 , \tag{2.27}$$

This property is the reason why Barnes' $G$-function appears in conformal field theory at $c = 1$. Due to the degenerate fusion rule (2.22), structure constants have nice properties under integer shifts of momentums, so they are expressed as $G$-functions of momentums. The reason why such integer shifts should produce $\Gamma$ functions is because monodromies of BPZ equations for correlation functions of degenerate fields are $\Gamma$ functions of momentums.

Barnes' $G$-function also obeys the duplication formula

$$G(2x + 1) = \frac{2^{2x^2}}{(2\pi)^x G(\frac{1}{2})G(\frac{3}{2})} \prod_{\pm,\pm} G\left(x + 1 \pm \tfrac{1}{4} \pm \tfrac{1}{4}\right) , \tag{2.28}$$

and the identities

$$G(k + \tfrac{3}{2}) \underset{k \in \mathbb{Z}}{=} (-)^{\frac{k(k+1)}{2}} \pi^{k + \frac{1}{2}} G(\tfrac{1}{2} - k) , \tag{2.29}$$

$$G(1 + 2r) \underset{r \in \frac{1}{4} + \frac{1}{2}\mathbb{Z}}{=} (-)^{2r^2 - \frac{1}{8}} \pi^{2r} G(1 - 2r) . \tag{2.30}$$

# 3 Compactified free boson

The massless compactified free boson at $c = 1$ is one of the simplest two-dimensional CFTs. It could even be called trivial, due to its abelian affine symmetry, although we will find a few subtleties with signs of correlation functions. The theory becomes less trivial if we forget about the affine symmetry, and consider it from the point of view of Virasoro symmetry only. In particular this requires us to compute correlation functions of Virasoro primary fields that are not affine primary fields. This will serve as a warm-up for the Ashkin–Teller model.

## 3.1 Chiral properties

**Current and primary fields**

The abelian affine symmetry algebra $\hat{\mathfrak{u}}_1$ is generated by a holomorphic current $J(z) = \sum_{n \in \mathbb{Z}} z^{-n-1} J_n$ obeying the following OPE and mode commutation relations

$$J(z)J(w) = \frac{1}{2(z-w)^2} + O(1) \qquad , \qquad [J_n, J_m] = \frac{n}{2} \delta_{n,-m} . \tag{3.1}$$

At $c = 1$, the energy-momentum tensor is the normal-ordered product of the current with itself, $T = (JJ)$. The generators $L_n$ of the corresponding Virasoro algebra are

$$L_{n \neq 0} = \sum_{m=-\infty}^{\infty} J_{n-m}J_m \qquad , \qquad L_0 = J_0^2 + 2\sum_{m=1}^{\infty} J_{-m}J_m . \tag{3.2}$$

An affine primary field, also known as a vertex operator, is defined by its OPE with the current,

$$J(z)V_{p,\bar{p}}(w) = \frac{p}{z-w}V_{p,\bar{p}}(w) + O(1) \ , \tag{3.3}$$

or equivalently by the action of affine algebra generators,

$$J_0 V_{p,\bar{p}} = p V_{p,\bar{p}} \qquad , \qquad J_{n>0} V_{p,\bar{p}} = 0 \ . \tag{3.4}$$

Using Eq. (3.2), we have $L_0 V_{p,\bar{p}} = J_0^2 V_{p,\bar{p}}$, and a vertex operator is also a Virasoro primary field, with the conformal dimension $\Delta = p^2$. This formally coincides with the definition (2.21) of the momentum, which we can therefore identify with the $\hat{\mathfrak{u}}_1$ charge.

## $\mathbb{Z}_2$ automorphism

The abelian affine Lie algebra has the automorphism $J \to -J$, which leaves the Virasoro algebra invariant. The corresponding action on the $\hat{\mathfrak{u}}_1$ charge is the conjugation

$$p^* = -p \ . \tag{3.5}$$

Two conjugate charges correspond to the same conformal dimension and therefore to the same Virasoro representation, although the affine representations differ.

## Representations and characters

For $p \notin \frac{1}{2}\mathbb{Z}$, the affine highest-weight representation generated by the affine primary field $V_p$ coincides with the Verma module of the Virasoro algebra with the conformal dimension $\Delta = p^2$. In particular, their characters agree,

$$\widehat{\chi}_p(q) = \chi_p(q) = \frac{q^{p^2}}{q^{\frac{1}{24}} \prod_{n=1}^{\infty}(1-q^n)} \ . \tag{3.6}$$

For $p \in \frac{1}{2}\mathbb{Z}$, this identity of characters still holds, but the underlying representations no longer coincide. For example, in the case $p = 0$, the Virasoro descendant field $L_{-1}V_0 = 2J_0 J_{-1}V_0$ vanishes. Therefore, the Virasoro representation generated by the affine primary field $V_0$ is not a Verma module, but a degenerate representation, i.e. the irreducible quotient of a Verma module by its maximal submodule. The affine descendant $J_{-1}V_0$ is absent from that Virasoro representation, and it generates another degenerate Virasoro representation. Actually, the affine representation that is generated by $V_k$ is an infinite sum of degenerate Virasoro representations [14]. The corresponding character identity reads

$$k \in \tfrac{1}{2}\mathbb{Z} \quad \Longrightarrow \quad \widehat{\chi}_k(q) = \sum_{k'=|k|}^{\infty} \chi_{k'}^{\text{degenerate}}(q) \ , \tag{3.7}$$

where the degenerate Virasoro characters are

$$k \in \tfrac{1}{2}\mathbb{N} \quad \Longrightarrow \quad \chi_k^{\text{degenerate}}(q) = \frac{q^{k^2} - q^{(k+1)^2}}{q^{\frac{1}{24}} \prod_{n=1}^{\infty}(1-q^n)} \ . \tag{3.8}$$

To summarize, the decomposition of the affine representation with momentum $k \in \frac{1}{2}\mathbb{Z}$ into Virasoro representations is given by character identity (3.7), rather than by the identity (3.6), which is true but misleading in this case.

## 3.2 Primary fields and fusion rules

**Primary fields and torus partition function**

The compactified free boson depends on a parameter $R \in \mathbb{C}^*$ called the compactification radius. The affine primary fields have $\hat{\mathfrak{u}}_1$ charges of the type

$$p = p_{(n,w)} = \frac{nR + wR^{-1}}{2} \quad , \quad \bar{p} = p_{(n,-w)} = \frac{nR - wR^{-1}}{2} \, , \qquad (n, w) \in \mathbb{Z} \, . \quad (3.9)$$

Depending on the context, the corresponding affine primary fields may be written as $V_{(n,w)} = V_{p_{(n,w)},p_{(n,-w)}} = V_{p_{(n,w)},\bar{p}_{(n,w)}}$. The conformal spin of a primary field is

$$S(V_{(n,w)}) = nw \in \mathbb{Z} \, . \quad (3.10)$$

Let us warn that the radius $R$ seldom appears explicitly in our formulas: most often, $R$ is hidden in a dependence on the charge $p$.

The modular invariant torus partition function reads

$$Z(q) = \sum_{(n,w) \in \mathbb{Z}^2} \widehat{\chi}_{p_{(n,w)}}(q) \widehat{\chi}_{p_{(n,-w)}}(\bar{q}) \, . \quad (3.11)$$

From the point of view of the Virasoro algebra, for any $(n, w) \neq (0, 0)$ we have two primary fields $V_{(n,w)}$ and $V_{(-n,-w)}$ with the same left and right conformal dimensions. Assuming the radius $R$ is generic, these fields generate Verma modules. On the other hand, the identity field $I = V_{0,0}$ generates a degenerate Virasoro representation. In the affine representation generated by $I$, there are infinitely many Virasoro primary fields $I_{k,\bar{k}}$ with $k, \bar{k} \in \mathbb{N}$.

Let us summarize the compactified free boson's primary fields:

| Sector | Affine primary fields | Virasoro primary fields |
|--------|----------------------|-------------------------|
| Identity | $I = I_{0,0}$ | $I_{k,\bar{k}} \quad \text{with} \quad k, \bar{k} \in \mathbb{N}$ |
| Vertex | $V_{(n,w)} \quad \text{with}$ | $\begin{cases} n, w \in \mathbb{Z} \\ (n, w) \neq (0, 0) \end{cases}$ |

$(3.12)$

**Fusion rules**

The affine fusion rules are dictated by conservation of momentum, and are simply

$$V_{(n_1,w_1)} \widehat{\times} V_{(n_2,w_2)} \sim V_{(n_1+n_2,w_1+w_2)} \, . \quad (3.13)$$

The Virasoro fusion rules for vertex operators are the same, except when we obtain the identity field $I_{0,0}$. In this case, the affine fusion rules omit infinitely many Virasoro primary fields that are affine descendant fields. These primary fields should however be written in the Virasoro fusion rules,

$$V_{(n,w)} \times V_{(-n,-w)} \sim \sum_{k,\bar{k}=0}^{\infty} I_{k,\bar{k}} \, . \quad (3.14)$$

To complete the Virasoro fusion rules of the model, we should also write fusion products that involve the Virasoro primary fields $I_{k,\bar{k}}$. By affine symmetry, these can be deduced from the trivial affine fusion product $I \widehat{\times} V_{(n,w)} \sim V_{(n,w)}$, which implies

$$I_{k,\bar{k}} \times V_{(n,w)} \sim V_{(n,w)} \, . \quad (3.15)$$

Comparing with the chiral fusion rule for degenerate fields (2.22), we observe that only one of the allowed $(2k+1)(2\bar{k}+1)$ terms is present. With a generic radius $R$, the other terms would not have momentums of the type $(p, \bar{p}) = (p_{(n,w)}, p_{(n,-w)})$ (3.9).

In the fusion product $I_{k_1} \times I_{k_2}$ of two chiral degenerate fields, the Virasoro fusion rules (2.23) would lead to $\min(2k_1 + 1, 2k_2 + 1)$ terms. However, we will now show that affine symmetry forbids some of these terms, due to the $\mathbb{Z}_2$ symmetry. As an affine descendant of the identity, $I_k$ must be even or odd under $\mathbb{Z}_2$ symmetry $J \to -J$, for example $I_1 \propto J_{-1}V_0$ is odd. In the Virasoro fusion rule $I_1 \times I_1 \sim I_0 + I_1 + I_2$, we now see that the second term must drop out, and that $I_2$ must be even. Iterating, we find the $\mathbb{Z}_2$ conjugation rule

$$\boxed{I_k^* = (-)^k I_k} \ , \tag{3.16}$$

and the fusion rules

$$\boxed{I_{k_1, \bar{k}_1} \times I_{k_2, \bar{k}_2} \sim \sum_{k \overset{2}{=} |k_1 - k_2|}^{k_1 + k_2} \sum_{\bar{k} \overset{2}{=} |\bar{k}_1 - \bar{k}_2|}^{\bar{k}_1 + \bar{k}_2} I_{k, \bar{k}}} \ . \tag{3.17}$$

Here the sums run by increments of 2, whereas Virasoro symmetry would allow increments of 1. For basic checks of the disappearance of some terms allowed by Virasoro symmetry, see Appendix A.2.

## 3.3   Correlation functions of affine primary fields

### $N$-point functions

Due to affine symmetry, $N$-point functions of affine primary fields should behave as

$$\left\langle \prod_{i=1}^{N} V_{(n_i, w_i)}(z_i) \right\rangle \propto \delta_{\sum n_i, 0} \delta_{\sum w_i, 0} \prod_{1 \le i < j \le N} \left| z_{ij}^{2p_i p_j} \right|^2 \ , \tag{3.18}$$

where we use the notation (2.5), and where the proportionality coefficient should be $z_i$-independent. In this expression, the differences between the left and right exponents are

$$2p_i p_j - 2\bar{p}_i \bar{p}_j = n_i w_j + n_j w_i \ . \tag{3.19}$$

Since these are integers, our expression is single-valued. However, since these integers are not necessarily even, our expression is not manifestly invariant under field permutations. To make it invariant, we should choose an appropriate $z_i$-independent prefactor. We propose the permutation-invariant expression

$$\left\langle \prod_{i=1}^{N} V_{(n_i, w_i)}(z_i) \right\rangle = \delta_{\sum n_i, 0} \delta_{\sum w_i, 0} \prod_{i < j} (-)^{w_i n_j} \left| z_{ij}^{2p_i p_j} \right|^2 \ . \tag{3.20}$$

### Structure constants

Let us focus on the cases $N = 2, 3$, and deduce the two- and three-point structure constants. In the case $N = 2$, we have

$$\left\langle V_{(n,w)}(z_1) V_{(-n,-w)}(z_2) \right\rangle = \frac{(-)^{nw}}{\left| z_{12}^{2\Delta} \right|^2} \ . \tag{3.21}$$

Comparing with the general two-point function (2.4), we deduce the conjugation relation

$$(n, w)^* = (-n, -w) \ , \tag{3.22}$$

and the two-point structure constant

$$B_{(n,w)} = (-)^{nw} \ . \tag{3.23}$$

In the case $N = 3$, we compare with the general formula (2.6) and obtain the three-point structure constant $\prod_{i<j}(-)^{w_i n_j}$. Using momentum conservation, we rewrite this in a way that is manifestly invariant under cyclic permutations,

$$C_{(n_1, w_1)(n_2, w_2)(n_3, w_3)} = (-)^{n_1 w_2 + n_2 w_3 + n_3 w_1} \ , \tag{3.24}$$

It is straightforward to check that it also has the expected behaviour under odd permutations (2.7). Moreover, the relation $C_{(0,0)(n,w)(-n,-w)} = B_{(n,w)}$ is compatible with $V_{(0,0)}$ being the identity field. The three-point function therefore reads

$$\left\langle \prod_{i=1}^{3} V_{(n_i, w_i)}(z_i) \right\rangle = (-)^{n_1 w_2 + n_2 w_3 + n_3 w_1} \frac{\delta_{n_1 + n_2 + n_3, 0} \delta_{w_1 + w_2 + w_3, 0}}{\left| z_{12}^{\Delta_1 + \Delta_2 - \Delta_3} z_{23}^{\Delta_2 + \Delta_3 - \Delta_1} z_{13}^{\Delta_3 + \Delta_1 - \Delta_2} \right|^2} \ . \tag{3.25}$$

**Crossing symmetry**

The symmetry of our four-point function under field permutations guarantees that it is crossing symmetric. Let us demonstrate how this works at the level of structure constants and conformal blocks.

Due to momentum conservation, any four-point function of vertex operators contains only one affine conformal block in any channel. Using our three-point structure constants (3.24), we compute the four-point structure constants (2.12) and find

$$D_{1+2|1234}^{(s)} = \frac{C_{1,2,-1-2} C_{1+2,3,4}}{B_{1+2}} = (-)^{\sum_{i<j} w_i n_j} \tag{3.26}$$

where we have used a schematic notation $V_{1+2}$ for the unique field in the fusion of $V_1$ and $V_2$. The affine $s$-channel conformal block is

$$\widehat{\mathcal{F}}_{1+2|1234}^{(s)} = \prod_{i<j} z_{ij}^{2p_i p_j} \ , \tag{3.27}$$

where the $z_i$-independent prefactor is determined by the normalization condition (2.13). To get $t$-channel blocks and structure constants from $s$-channel expressions, it is enough to permute the fields $V_1$ and $V_3$, and we find

$$D_{2+3|1234}^{(t)} = D_{2+3|3214}^{(s)} = (-)^{S_1 + S_2 + S_3} D_{1+2|1234}^{(s)} \ , \tag{3.28}$$

$$\left| \widehat{\mathcal{F}}_{1+2|1234}^{(t)} \right|^2 = \left| \widehat{\mathcal{F}}_{2+3|3214}^{(s)} \right|^2 = (-)^{S_1 + S_2 + S_3} \left| \widehat{\mathcal{F}}_{1+2|1234}^{(s)} \right|^2 \ . \tag{3.29}$$

This leads to the rather trivial crossing symmetry equation

$$D_{1+2|1234}^{(s)} \left| \widehat{\mathcal{F}}_{1+2|1234}^{(s)} \right|^2 = D_{2+3|1234}^{(t)} \left| \widehat{\mathcal{F}}_{1+2|1234}^{(t)} \right|^2 \ . \tag{3.30}$$

## 3.4 Correlation functions of Virasoro primary fields

After solving the free boson theory in terms of the affine symmetry algebra, let us now consider the same problem in terms of the Virasoro algebra alone. The difference is only visible when the Virasoro representations differ from the affine representations, which for generic $R$ happens only in the vacuum sector.

In principle, the correlation functions of Virasoro primary fields can be deduced from those of affine primary fields using affine Ward identities. In practice, in order to derive explicit formulas for correlation functions of arbitrary Virasoro primary fields, we will solve crossing symmetry equations for four-point functions that involve a degenerate field of momentum $k = 1$. The affine symmetry ensures that the equations are chirally factorized, and we will solve chiral crossing symmetry equations, whose solutions are chiral structure constants.

### Structure constants from the affine algebra

Let us normalize the Virasoro primary fields in the vacuum sector $I_{k,\bar{k}}$ such that their two-point functions are one,

$$B_{I_{k,\bar{k}}} = \left\langle I_{k,\bar{k}} I_{k,\bar{k}} \right\rangle = 1 \ . \tag{3.31}$$

This defines $I_{k,\bar{k}}$ up to a sign ambiguity $I_{k,\bar{k}} \to -I_{k,\bar{k}}$, which we will later fix. The field $I_{k,\bar{k}}$ is an affine descendant of the identity field $I_{0,0}$, so its correlation functions can in principle be computed using the affine symmetry. Let $\mathcal{O}_k$ be the affine creation operators such that $I_{k,\bar{k}} = \mathcal{O}_k \bar{\mathcal{O}}_{\bar{k}} I_{0,0}$, then $\frac{\left\langle I_{k,\bar{k}} V_{p,\bar{p}} V_{-p,-\bar{p}} \right\rangle}{\left\langle I_{0,0} V_{p,\bar{p}} V_{-p,-\bar{p}} \right\rangle}$ factorizes into a chiral and an anti-chiral factor, and we can write

$$\boxed{\left\langle I_{k,\bar{k}} V_{p,\bar{p}} V_{-p,-\bar{p}} \right\rangle = f_{k,p} f_{\bar{k},\bar{p}} B_{V_{p,\bar{p}}}} \ . \tag{3.32}$$

We will refer to the chiral factor $f_{k,p}$ as a chiral structure constant. By definition, this factor obeys $f_{0,p} = 1$. Similarly, we define chiral structure constants $g_{k_1,k_2,k_3}$ for the three-point functions of degenerate fields,

$$\boxed{\left\langle I_{k_1,\bar{k}_1} I_{k_2,\bar{k}_2} I_{k_3,\bar{k}_3} \right\rangle = \left| \delta_{k_1+k_2+k_3 \in 2\mathbb{Z}} \, g_{k_1,k_2,k_3} \right|^2} \ . \tag{3.33}$$

A few chiral structure constants can be directly computed from affine Ward identities. To do this, we should first determine the expressions $I_{k,\bar{k}} = \mathcal{O}_k \bar{\mathcal{O}}_{\bar{k}} I_{0,0}$ of degenerate fields as affine descendant fields. The affine creation operator $\mathcal{O}_k$ is determined up to a sign by requiring that $\mathcal{O}_k I_{0,0}$ be a Virasoro primary field of dimension $k^2$, normalized such that its two-point function is one. We find

$$I_{1,0} = \sqrt{2} J_{-1} I_{0,0} \ , \tag{3.34}$$

$$I_{2,0} = -\frac{1}{3\sqrt{6}} \left( 3J_{-2}^2 - 4J_{-3}J_{-1} + 4J_{-1}^4 \right) I_{0,0} \ , \tag{3.35}$$

$$
\begin{aligned}
I_{3,0} = \frac{1}{3\sqrt{5}} \Bigg( &-\frac{4}{45} J_{-1}^9 + \frac{8}{15} J_{-3} J_{-1}^6 - \frac{2}{5} J_{-2}^2 J_{-1}^5 - \frac{4}{5} J_{-5} J_{-1}^4 + 2 J_{-4} J_{-2} J_{-1}^3 \\
&- 2 J_{-3} J_{-2}^2 J_{-1}^2 + \frac{3}{4} J_{-2}^4 J_{-1} - \frac{3}{4} J_{-4}^2 J_{-1} + \frac{4}{5} J_{-5} J_{-3} J_{-1} \\
&- \frac{4}{9} J_{-3}^3 - \frac{3}{5} J_{-5} J_{-2}^2 + J_{-4} J_{-3} J_{-2} \Bigg) I_{0,0} \ .
\end{aligned}
\tag{3.36}
$$

Notice that these expressions are compatible with the behaviour (3.16) of $I_{k,0}$ under the $\mathbb{Z}_2$ symmetry: $I_{1,0}, I_{3,0}$ are $\mathbb{Z}_2$-odd while $I_{2,0}$ is $\mathbb{Z}_2$-even. Using the affine Ward identity

$$\left\langle J_{-1}V_{p_1,\bar{p}_1}(z_1)V_{p_2,\bar{p}_2}(z_2)V_{p_3,\bar{p}_3}(z_3)\right\rangle = \left(\frac{p_2}{z_{12}} + \frac{p_3}{z_{13}}\right)\left\langle\prod_{i=1}^{3}V_{p_i,\bar{p}_i}(z_i)\right\rangle , \tag{3.37}$$

we compute

$$\left\langle I_{1,0}V_{p,\bar{p}}V_{-p,-\bar{p}}\right\rangle = \sqrt{2}pB_{V_{p,\bar{p}}} \qquad \text{so that} \qquad f_{1,p} = \sqrt{2}p . \tag{3.38}$$

Using more complicated affine Ward identities, we compute

$$f_{2,p} = -\frac{1}{3\sqrt{6}}p^2(4p^2 - 1) , \tag{3.39}$$

$$g_{1,1,2} = \sqrt{\frac{2}{3}} . \tag{3.40}$$

Such pedestrian calculations are however not feasible for arbitrary degenerate fields $I_{k,\bar{k}}$, and we will use another approach.

## Structure constants from degenerate crossing symmetry equations

Let us work out crossing symmetry equations for four-point functions with one degenerate field of momentum $k = 1$ i.e. conformal dimension $\Delta = 1$. This is the simplest nontrivial degenerate field that appears in the free boson CFT and in the Ashkin–Teller model.

Of course, we should in principle also check the crossing symmetry equations for four-point functions with arbitrary degenerate fields. However, all the other degenerate fields can be obtained from our $k = 1$ degenerate field, by repeatedly fusing it with itself. Therefore, their crossing symmetry equations are in principle consequences of the crossing symmetry equations that we will study.

In order to determine the chiral structure constant $f_{k,p}$, we solve the crossing symmetry equations (2.20) for the chiral four-point function $\langle I_1 V_p V_p I_k\rangle$, which involves the fusion rule (3.15) and the chiral structure constant $f_{k,p}$ (3.32):

$$f_{1,p}f_{k,p}F_{0,\epsilon}\begin{bmatrix} p & p \\ 1 & k \end{bmatrix} = \begin{cases} g_{1,k,k+\epsilon}f_{k+\epsilon,p} & \text{if } \epsilon \in \{+, -\} , \\ 0 & \text{if } \epsilon = 0 , \end{cases} \tag{3.41}$$

where $g_{1,k,k+\epsilon}$ will shortly be determined, see Eq. (3.51). The fusion kernels are given in Eq. (5.46), and we find the solution

$$\boxed{f_{k,p} = 2^{-k}(-1)^{\frac{k(k-1)}{2}}\sqrt{\frac{\Gamma(\frac{1}{2})\Gamma(1+k)}{\Gamma(\frac{1}{2}+k)}\frac{G(1+k)^2}{G(1+2k)}}\prod_{\pm}\frac{G(1+2p\pm k)}{G(1+2p)}} . \tag{3.42}$$

This is actually a polynomial function of $p$, as is manifest in the equivalent expression

$$f_{k,p} = (-1)^{\frac{k(k-1)}{2}}\sqrt{\frac{\Gamma(\frac{1}{2})\Gamma(1+k)}{\Gamma(\frac{1}{2}+k)}\frac{G(1+k)^2}{G(1+2k)}}p^k\prod_{i=1}^{k-1}(4p^2 - i^2)^{k-i} . \tag{3.43}$$

Special cases of this formula include

$$f_{0,p} = 1 , \tag{3.44}$$

$$f_{1,p} = \sqrt{2}p , \tag{3.45}$$

$$f_{2,p} = -\frac{1}{3\sqrt{6}}p^2(4p^2 - 1) . \tag{3.46}$$

In particular, this agrees with the determination (3.39) of $f_{2,p}$ from affine symmetry. This agreement is only significant up to an overall sign, as the definition of our Virasoro primary field $I_{2,0}$ (3.35) allows us to flip its sign.

Moreover, $f_{k,p}$ obeys

$$f_{k,-p} = (-)^k f_{k,p} \,, \tag{3.47}$$

which is necessary for the three-point structure constant (3.32) to have the right behaviour under field permutations, since $(-)^{\Delta_k} = (-)^{k^2} = (-)^k$. The zeros of $f_{k,p}$ for $p \in \{0, \frac{1}{2}, \ldots, \frac{k-1}{2}\}$ are due to the affine primary field $V_p$ becoming a degenerate Virasoro primary field $V_p \propto I_p$, such that the fusion rule $I_p \times I_p$ (2.22) does not include $I_k$ if $k > 2p$.

For a given value of the radius $R$, the momentum $p$ takes discrete values $p_{n,w}$ (3.9). However, the chiral structure constants $f_{k,p}$ only depend on $R, n, w$ through their polynomial dependence on $p$.

### Case with three degenerate fields

In order to determine the chiral three-point structure constants $g_{k_1,k_2,k_3}$ (3.33), we use the crossing symmetry equation (2.20) for the chiral four-point function $\langle I_1 I_{k_1} I_{k_2} I_{k_3} \rangle$, which involves the fusion rule (3.17) and the chiral structure constant $g_{k_1,k_2,k_3}$ (3.33):

$$\sum_{\epsilon_1 = \pm} g_{1,k_1,k_1+\epsilon_1} g_{k_1+\epsilon_1,k_2,k_3} F_{\epsilon_1,\epsilon_3} \begin{bmatrix} k_1 & k_2 \\ 1 & k_3 \end{bmatrix} = \begin{cases} g_{1,k_3,k_3+\epsilon_3} g_{k_1,k_2,k_3+\epsilon_3} & \text{if } \epsilon_3 \in \{+,-\} \,, \\ 0 & \text{if } \epsilon_3 = 0 \,. \end{cases} \tag{3.48}$$

The relevant fusion kernels are given by Eq. (5.44). We derive the solution for $g_{k_1,k_2,k_3}$ from the second crossing symmetry equation, and check it using the first crossing symmetry equation. The result is

$$\boxed{g_{k_1,k_2,k_3} = \frac{G(1+k_{123})\Gamma(1+\frac{1}{2}k_{123})\widetilde{G}(1+k_{12}^3)\widetilde{G}(1+k_{13}^2)\widetilde{G}(1+k_{23}^1)}{\prod_{i=1}^3 G(1+2k_i)\sqrt{\Gamma(\frac{1}{2})\Gamma(\frac{1}{2}+k_i)\Gamma(1+k_i)}}} \,, \tag{3.49}$$

where we introduce the notations $k_{12}^3 = k_1 + k_2 - k_3$ and $k_{123} = k_1 + k_2 + k_3$, as well as

$$\widetilde{G}(x) = G(x)\Gamma(\tfrac{x}{2}) \quad \text{so that} \quad \frac{\widetilde{G}(x+1)}{\widetilde{G}(x-1)} = \frac{1}{2}\Gamma(x)^2 \,. \tag{3.50}$$

The chiral three-point structure constant is normalized so that $g_{0,k,k} = 1$. And we have the special case

$$g_{1,k,k+1} = \sqrt{\frac{k+1}{2k+1}} \,. \tag{3.51}$$

## 4  Ashkin–Teller model

### 4.1  Space of states and primary fields

The Ashkin–Teller model is a $\mathbb{Z}_2$ orbifold of the compactified free boson, where the $\mathbb{Z}_2$ is the diagonal action of the affine algebra's automorphism,

$$(J, \bar{J}) \to (J, \bar{J})^* = (-J, -\bar{J}) \,. \tag{4.1}$$

As a result, the symmetry algebra of the orbifold CFT include combinations such as $J^2, J\bar{J}, \bar{J}^2$, but not the individual chiral currents $J, \bar{J}$.

## Untwisted sector

For vertex operators $V_{p,\bar{p}}$ with $(p, \bar{p}) \neq (0, 0)$, the effect of the orbifold is just the identification $V_{p,\bar{p}} = V_{-p,-\bar{p}}$. The structure of the affine Verma modules is not affected. For example, to see that the level one descendant state survives, just write it as $J_0 J_{-1} V_{p,-\bar{p}}$ rather than $J_{-1} V_{p,-\bar{p}}$. The situation is more complicated in the vacuum sector, where $J_0 V_{0,0} = 0$ and $J_{-1} V_{0,0}$ is projected out. The behaviour of a Virasoro primary field $I_{k,\bar{k}}$ under the action $\mathbb{Z}_2$ can be deduced from Eq. (3.16), and it follows that $I_{k,\bar{k}}$ survives if and only if $k - \bar{k} \in 2\mathbb{Z}$, in other words if $I_{k,\bar{k}}$ has even conformal spin. This determines the structure of the orbifold's vacuum sector.

## Twisted sector

Together, the orbifold's vacuum and vertex sectors constitute the untwisted sector. From the torus partition function, it can be seen that the theory must also include a twisted sector, generated by two affine primary fields called twist fields. We denote these fields as $T^\epsilon$ with $\epsilon = 0, 1 \mod 2$. By definition, the monodromy of the currents $J, \bar{J}$ around a twist field amounts to the action of $\mathbb{Z}_2$, and we have the OPE

$$
J(z) T^\epsilon(w) = \frac{1}{\sqrt{z - w}} J_{-\frac{1}{2}} T^\epsilon(w) + O\left(\sqrt{z - w}\right) . \tag{4.2}
$$

In the twist sector, the current is half-integer moded, $J(z) = \sum_{n \in \mathbb{Z} + \frac{1}{2}} z^{-n-1} J_n$. Twist fieds are affine primary fields in the sense that

$$
J_{n \in \mathbb{N} + \frac{1}{2}} T^\epsilon = 0 . \tag{4.3}
$$

And the twist representation of the abelian affine Lie algebra is built by acting on a twist field with the half-integer moded creation operators $J_{-\frac{1}{2}}, J_{-\frac{3}{2}}, \cdots$. The commutation relations between these modes are the same as in the untwisted sector (3.1), but with half-integer values of the indices. Virasoro generators in the twisted sector are given by

$$
L_{n \neq 0} = \sum_{m = \frac{1}{2} + \mathbb{Z}} J_{n-m} J_m \quad , \qquad L_0 = \frac{1}{16} + 2 \sum_{m \in \frac{1}{2} + \mathbb{N}} J_{-m} J_m . \tag{4.4}
$$

Therefore, the twist fields $T^\epsilon$ have the conformal dimensions $(\Delta, \bar{\Delta}) = (\frac{1}{16}, \frac{1}{16})$.

Just like the affine identity representation, the affine twist representation is reducible as a representation of the Virasoro algebra, as it contains Virasoro primary fields that are affine descendants. These are called higher twist fields. The first examples are [5]

$$
T^\epsilon_{\frac{3}{4}, \frac{1}{4}} = 2 J_{-\frac{1}{2}} T^\epsilon , \tag{4.5}
$$

$$
T^\epsilon_{\frac{5}{4}, \frac{1}{4}} = \frac{2}{3} \left( J_{-\frac{3}{2}} - 4 J^3_{-\frac{1}{2}} \right) T^\epsilon , \tag{4.6}
$$

where we label higher twist fields by their left and right momentums. More generally, there is one higher twist field for each pair of momentums $(r, \bar{r}) \in \left(\frac{1}{4} + \frac{1}{2}\mathbb{N}\right)^2$. However, not all these higher twist fields belong to the orbifold theory. Under the action (4.1) of $\mathbb{Z}_2$, a twist field picks a sign that depends on the number (even or odd) of modes of $J, \bar{J}$ in its expression as a descendant of $T^\epsilon$. Since each $J$-mode adds a half-integer to the left conformal dimension, the parity of the number of $J$-modes can be deduced from the difference between the left conformal dimension of the higher twist field and the

left conformal dimension of a lowest twist field, schematically $(-)^{\#J} = (-)^{2\left(\Delta - \frac{1}{16}\right)}$ and similarly $(-)^{\#\bar{J}} = (-)^{2\left(\bar{\Delta} - \frac{1}{16}\right)}$. We therefore find that twist fields behave as

$$\left(T^\epsilon_{r,\bar{r}}\right)^* = (-)^{2r^2 - 2\bar{r}^2} T^\epsilon_{r,\bar{r}} \ . \tag{4.7}$$

Here the combination $r^2 - \bar{r}^2$ is the conformal spin, which can take integer or half-integer values. Projecting on $\mathbb{Z}_2$-invariant states therefore amounts to selecting states with integer spins. (Compare with the identity sector, where spins of degenerate fields were integer to start with, and the projection selected even spins.) Let us list the first few non-chiral twist fields with integer spins, ordered by their total conformal dimension $\Delta + \overline{\Delta}$:

$$T^\epsilon_{\frac{1}{4},\frac{1}{4}} = T^\epsilon \ , \quad T^\epsilon_{\frac{3}{4},\frac{3}{4}} \ , \quad T^\epsilon_{\frac{5}{4},\frac{3}{4}} \ , \quad T^\epsilon_{\frac{5}{4},\frac{5}{4}} \ , \quad T^\epsilon_{\frac{7}{4},\frac{1}{4}} \ , \quad T^\epsilon_{\frac{9}{4},\frac{1}{4}} \ , \dots \tag{4.8}$$

**Summary**

Let us summarize the primary fields of the Ashkin–Teller model, which can be compared with the primary fields (3.12) of the compactified free boson:

| Sector | Affine primary fields | Virasoro primary fields | |
|---|---|---|---|
| Identity | $I = I_{0,0}$ | $I_{k,\bar{k}}$ with $\begin{cases} k, \bar{k} \in \mathbb{N} \\ k - \bar{k} \in 2\mathbb{Z} \end{cases}$ | |
| Vertex | $V_{(n,w)} = V_{(-n,-w)}$ | $V_{(n,w)}$ with $\begin{cases} n, w \in \mathbb{Z} \\ (n,w) \neq (0,0) \end{cases}$ | (4.9) |
| Twist | $T^\epsilon = T^\epsilon_{\frac{1}{4},\frac{1}{4}}$ | $T^\epsilon_{r,\bar{r}}$ with $\begin{cases} \epsilon = 0, 1 \bmod 2 \\ r, \bar{r} \in \frac{1}{4} + \frac{1}{2}\mathbb{N} \\ r^2 - \bar{r}^2 \in \mathbb{Z} \end{cases}$ | |

This agrees with the known torus partition function [14, 3, 15].

## 4.2 Fusion rules

**Vertex operators**

The affine fusion rules of vertex operators are obtained from the free boson fusion rules (3.13) by taking the action of $\mathbb{Z}_2$ into account,

$$V_{(n_1,w_1)} \widehat{\times} V_{(n_2,w_2)} \sim V_{(n_1+n_2,w_1+w_2)} + V_{(n_1-n_2,w_1-w_2)} \ . \tag{4.10}$$

In words, momentum is now conserved only up to a sign. The Virasoro fusion rules are the same as the affine fusion rules, except when the identity sector is involved, which happens if $(n_1, w_1) = \pm(n_2, w_2)$. In that case, we obtain additionally a sum over all Virasoro primary fields in the identity sector,

$$\boxed{V_{(n,w)} \times V_{(n,w)} \sim V_{(2n,2w)} + \sum_{\substack{k,\bar{k} \in \mathbb{N} \\ k-\bar{k} \in 2\mathbb{Z}}} I_{k,\bar{k}}} \ . \tag{4.11}$$

The trivial fusion rule $I_{k,\bar{k}} \times V_{(n,w)} \sim V_{(n,w)}$ (3.15) is the same in the orbifold as in the free boson, and the orbifold's restriction $k - \bar{k} \in 2\mathbb{Z}$ does not affect the fusion rules (3.17) of the identity sector.

**Twist fields**

When it comes to twist fields, affine fusion rules can be deduced from the four-point functions (4.25) determined by Al. Zamolodchikov [2],

$$T^{\epsilon_1} \widehat{\times} T^{\epsilon_2} \sim \frac{1}{2} \sum_{\substack{n \in \mathbb{Z} \\ w \in 2\mathbb{Z} + \epsilon_1 + \epsilon_2 \\ (n,w) \neq (0,0)}} V_{(n,w)} + \delta_{\epsilon_1, \epsilon_2} I \ . \tag{4.12}$$

Here the prefactor $\frac{1}{2}$ avoids counting the same field $V_{(n,w)} = V_{(-n,-w)}$ twice. By permutation symmetry of the fusion multiplicities, we can deduce the fusion rules

$$T^{\epsilon} \widehat{\times} V_{(n,w)} \sim T^{\epsilon+w} \ . \tag{4.13}$$

Now let us work out the fusion rules for Virasoro primary fields in the twist sector. This is most simple in the case

$$\boxed{T_{r,\bar{r}}^{\epsilon} \times V_{(n,w)} \sim \sum_{\substack{r', \bar{r}' \in \frac{1}{4} + \frac{1}{2}\mathbb{N} \\ r'^2 - \bar{r}'^2 \in \mathbb{Z}}} T_{r',\bar{r}'}^{\epsilon+w}} \ . \tag{4.14}$$

The fusion of a degenerate field with a higher twist field is deduced from the fusion of the identity field with the same higher twist field, and should therefore be chirally factorized. For example, the chiral degenerate field $I_1 = J$ has a well-known OPE (4.2) with the twist field $T^{\epsilon}$, from which we deduce the chiral fusion rule $I_1 \times T_{\frac{1}{4}} \sim T_{\frac{3}{4}} + T_{\frac{5}{4}}$. Notice the absence of the term $T_{\frac{1}{4}}$, which is allowed by Virasoro symmetry (2.22) but forbidden by affine symmetry. This leads to the fusion rules

$$\boxed{I_{k,\bar{k}} \times T_{r,\bar{r}}^{\epsilon} \sim \sum_{i \overset{2}{=} -k}^{k} \sum_{\bar{i} \overset{2}{=} -\bar{k}}^{\bar{k}} T_{|r+i|,|\bar{r}+\bar{i}|}^{\epsilon}} \ . \tag{4.15}$$

Notice that a given field $T_{|r+i|,|\bar{r}+\bar{i}|}^{\epsilon}$ can never appear twice, because $|r+i| = |r+i'| \implies i = i'$ since $r \notin \frac{1}{2}\mathbb{N}$.

By permutation symmetry of fusion multiplicities, we can deduce the fusion product of two higher twist fields, and unpack the identity sector in the affine fusion rule (4.12):

$$\boxed{T_{r_1,\bar{r}_1}^{\epsilon_1} \times T_{r_2,\bar{r}_2}^{\epsilon_2} \sim \frac{1}{2} \sum_{\substack{n \in \mathbb{Z} \\ w \in 2\mathbb{Z} + \epsilon_1 + \epsilon_2 \\ (n,w) \neq (0,0)}} V_{(n,w)} + \delta_{\epsilon_1, \epsilon_2} \sum_{k \overset{2}{=} |r_1 \pm_{\mathbb{Z}} r_2|}^{\infty} \sum_{\bar{k} \overset{2}{=} |\bar{r}_1 \pm_{\mathbb{Z}} \bar{r}_2|}^{\infty} I_{k,\bar{k}}} \ , \tag{4.16}$$

where we define the notation $r_1 \pm_{\mathbb{Z}} r_2$ for $r_1, r_2 \in \frac{1}{4} + \frac{1}{2}\mathbb{Z}$ by

$$\{r_1 + r_2, r_1 - r_2\} = \{r_1 \pm_{\mathbb{Z}} r_2, r_1 \mp_{\mathbb{Z}} r_2\} \quad \text{with} \quad \begin{cases} r_1 \pm_{\mathbb{Z}} r_2 \in \mathbb{Z} \ , \\ r_1 \mp_{\mathbb{Z}} r_2 \in \mathbb{Z} + \frac{1}{2} \ . \end{cases} \tag{4.17}$$

For example, $\frac{1}{4} \pm_{\mathbb{Z}} \frac{3}{4} = 1$ and $\frac{1}{4} \pm_{\mathbb{Z}} \frac{5}{4} = -1$. Notice that $r^2 - \bar{r}^2 \in \mathbb{Z} \implies r \pm_{\mathbb{Z}} \bar{r} \in 2\mathbb{Z}$, and this implies $|r_1 \pm_{\mathbb{Z}} r_2| - |\bar{r}_1 \pm_{\mathbb{Z}} \bar{r}_2| \in 2\mathbb{Z}$ in Eq. (4.16), so that $I_{k,\bar{k}}$ has even spin as it should. For some basic checks of these fusion rules, see Appendix A.2.

## 4.3 Correlation functions of affine primary fields

We will mostly focus on four-point functions, as this is sufficient for proving consistency of the model, and checking the three-point structure constants. Some higher-point correlation functions were computed by Al. Zamolodchikov [16].

**Vertex operators**

There is a simple recipe for deducing correlation functions of vertex operators in the orbifold from their free bosonic counterparts, based on the identifications

$$V^{\text{orbifold}}_{(n,w)} \underset{(n,w)\neq(0,0)}{=} \frac{1}{\sqrt{2}} \left( V^{\text{free boson}}_{(n,w)} + V^{\text{free boson}}_{(-n,-w)} \right) \quad , \quad V^{\text{orbifold}}_{(0,0)} = V^{\text{free boson}}_{(0,0)} \ . \tag{4.18}$$

Only in this equation do we use superscripts for distinguishing the two CFTs: in the rest of Section 4, vertex operators belong to the orbifold CFT. The numerical prefactors ensure that the orbifold vertex operators are normalized such that their two-point structure constants are the same as in the free boson (3.23). The orbifold three-point structure constants are then

$$C_{(n_1,w_1)(n_2,w_2)(n_3,w_3)} = \begin{cases} \frac{1}{\sqrt{2}}(-)^{n_1 w_2 + n_2 w_3 + n_3 w_1} & \text{if } \forall i, \ (n_i, w_i) \neq (0,0) \ , \\ 1 & \text{else} \ , \end{cases} \tag{4.19}$$

where the only difference with the free boson's structure constants (3.24) is the prefactor $\frac{1}{\sqrt{2}}$. For an orbifold $N$-point function $\left\langle \prod_{i=1}^{N} V_{(n_i,w_i)}(z_i) \right\rangle$, momentum conservation reads

$$\exists (\eta_i) \in \{+,-\}^N \ , \quad \sum_{i=1}^{N} \eta_i n_i = \sum_{i=1}^{N} \eta_i w_i = 0 \ , \tag{4.20}$$

and the orbifold $N$-point function reads

$$\left\langle \prod_{i=1}^{N} V_{(n_i,w_i)}(z_i) \right\rangle = \frac{(-)^{\sum_{i<j} w_i n_j}}{2^{\frac{N}{2}}} \sum_{(\eta_i)\in\{+,-\}^N} \delta_{(\sum \eta_i n_i, \sum \eta_i w_i),(0,0)} \prod_{i<j} \left| z_{ij}^{2\eta_i \eta_j p_i p_j} \right|^2 \ . \tag{4.21}$$

In particular, let us focus on four-point functions. There are three inequivalent cases:

- For generic $(n_i, w_i)$ such that $\sum n_i = \sum w_i = 0$, we have 2 solutions $(\eta_i) \in \{(+,+,+,+),(-,-,-,-)\}$, and the four-point function reads

$$\left\langle \prod_{i=1}^{4} V_{(n_i,w_i)}(z_i) \right\rangle = \frac{(-)^{\sum_{i<j} w_i n_j}}{2} \prod_{i<j} \left| z_{ij}^{2\eta_i \eta_j p_i p_j} \right|^2 \ . \tag{4.22}$$

In any channel, this is just one conformal block, with a numerical prefactor that comes from the two- and three-point structure constants (4.19).

- If the four fields coincide pairwise, we have 4 solutions for the signs $(\eta_i)$, and we obtain

$$\left\langle V_{(n,w)}(z_1) V_{(n,w)}(z_2) V_{(n',w')}(z_3) V_{(n',w')}(z_4) \right\rangle$$
$$= \frac{(-)^{nw+n'w'}}{2} \left| z_{12}^{-2p^2} z_{34}^{-2p'^2} \right|^2 \sum_{\pm} \left| \left( \frac{z_{13} z_{24}}{z_{14} z_{23}} \right)^{\pm 2pp'} \right|^2 \ . \tag{4.23}$$

In the $s$-channel, the sum of the two terms is interpreted as the contribution of the identity field $V_{(0,0)}$, with a prefactor $(-)^{nw+n'w'}$ that comes from the two-point structure constant (3.23), since the relevant three-point structure constant (4.19) is trivial. In the $t$- and $u$-channels, each term is the contribution of one of the two fields $V_{(n,w)\pm(n',w')}$.

- If the four fields all coincide, we have 6 solutions for the signs $(\eta_i)$, and we obtain

$$\left\langle \prod_{i=1}^{4} V_{(n,w)}(z_i) \right\rangle = \frac{\left| (z_{12}z_{34})^{4p^2} \right|^2 + \left| (z_{13}z_{24})^{4p^2} \right|^2 + \left| (z_{14}z_{23})^{4p^2} \right|^2}{2 \prod_{i<j} \left| z_{ij}^{2p^2} \right|^2} . \tag{4.24}$$

In the $s$-channel, the first term is the contribution of the field $V_{(2n,2w)}$, while the second and third terms are the contribution of the identity field $V_{(0,0)}$.

**Four-point function of twist fields**

The four-point function of twist fields was computed by Al. Zamolodchikov [2]. In our notations, it reads

$$\left\langle \prod_{i=1}^{4} T^{\epsilon_i} \right\rangle = \delta_{\sum \epsilon_i, 0} |F_0(q)|^2 \sum_{\substack{n \in \mathbb{Z} \\ w \in 2\mathbb{Z}+\epsilon_1+\epsilon_2}} (-)^{n(\epsilon_2+\epsilon_3)} q^{\Delta_{(n,w)}} \bar{q}^{\Delta_{(n,-w)}} , \tag{4.25}$$

where the conformal dimensions $\Delta_{(n,w)} = \frac{1}{4}(nR + wR^{-1})^2$ follow from Eqs. (2.21) and (3.9), $q$ is the nome associated to the cross-ratio of the four positions, and $F_0(q)$ is a known elliptic function of $q$. (See Appendix A.1 for details.) Interpreting this expression as an $s$-channel decomposition, we have already inferred the fusion rules (4.12), let us now infer the three-point structure constants. Since the relevant $s$-channel conformal blocks are $\mathcal{F}_\Delta^{(s)}(q) = F_0(q)(16q)^\Delta$ (A.26), the four-point structure constants must be

$$D_{(n,w)|\epsilon_1\epsilon_2\epsilon_3\epsilon_4}^{(s)} \underset{(n,w)\neq(0,0)}{=} 2(-)^{n(\epsilon_2+\epsilon_3)} 16^{-\Delta_{(n,w)}-\Delta_{(n,-w)}} , \tag{4.26}$$

where the prefactor 2 comes from grouping the contributions of $V_{(n,w)}$ and $V_{(-n,-w)}$. Given the two-point function of vertex operators (3.23), the four-point structure constants can be rewritten as combinations of two- and three-point structure constants (2.12), with the three-point structure constants

$$C_{\epsilon_1\epsilon_2(n,w)} \underset{(n,w)\neq(0,0)}{=} \sqrt{2}(-)^{n\epsilon_2} 4^{-\Delta_{(n,w)}-\Delta_{(n,-w)}} = \sqrt{2}(-)^{n\epsilon_2} 2^{-n^2R^2 - \frac{w^2}{R^2}} . \tag{4.27}$$

This expression has the right behaviour (2.7) under permutations, in particular

$$\frac{C_{\epsilon_1\epsilon_2(n,w)}}{C_{\epsilon_2\epsilon_1(n,w)}} = (-)^{n(\epsilon_1+\epsilon_2)} = (-)^{nw} . \tag{4.28}$$

In cases that involve the identity field $V_{(0,0)}$, three-point structure constants reduce to two-point structure constants, consistently with the triviality of conjugation and two-point structure constants in the twist sector,

$$\epsilon^* = \epsilon \quad , \quad B_\epsilon = 1 . \tag{4.29}$$

(This determines the two-point functions of twist fields via Eq. (2.4).)

To show that we have the correct three-point structure constants, it remains to check crossing symmetry of mixed four-point functions.

## Mixed correlation functions

As far as we know, four-point functions of two twist fields and two vertex operators have not been considered in the literature. In this case, we even have to determine the conformal blocks, see Appendix A. We consider the following four-point functions, and write the non-triviality condition according to the fusion rules,

$$\left\langle V_{(n_1,w_1)} V_{(n_2,w_2)} T^{\epsilon_2} T^{\epsilon_1} \right\rangle \neq 0 \quad \Longleftrightarrow \quad w_1 + w_2 + \epsilon_1 + \epsilon_2 \in 2\mathbb{Z} \ . \tag{4.30}$$

The $s$- and $t$-channel decompositions of these four-point functions are respectively predicted by the fusion rules (4.10) and (4.13), and schematically represented as follows:

$$\tag{4.31}$$

We denote the relevant $s$-channel conformal blocks as $\widehat{\mathcal{F}}^{(s)}_{1+2}(x), \widehat{\mathcal{F}}^{(s)}_{1-2}(x)$. From Eq. (A.9), we have

$$\widehat{\mathcal{F}}^{(s)}_{1+2}(x) = 16^{p_1 p_2}(1-x)^{-p_1^2} \left( \frac{1 - \sqrt{1-x}}{1 + \sqrt{1-x}} \right)^{2p_1 p_2} \ . \tag{4.32}$$

The chiral $t$-channel conformal block agrees with the $s$-channel conformal block up to a numerical prefactor. From Eq. (A.10), we have

$$\widehat{\mathcal{F}}^{(t)} = 16^{-p_1 p_2} \widehat{\mathcal{F}}^{(s)}_{1+2}(x) \ . \tag{4.33}$$

However, we know that the orbifold's twist representation is not chirally factorized: rather, it is obtained from the product of a left-moving and a right-moving twist representation, by projecting on states with integer spins. At the level of $t$-channel conformal blocks, integer spins correspond to integer powers of $1 - x$. Selecting integer powers of $1 - x$ in $\left|\widehat{\mathcal{F}}^{(t)}\right|^2$ amounts to combining it with its image under $\sqrt{1-x} \to -\sqrt{1-x}$, equivalently under $(p_2, \bar{p}_2) \to (-p_2, -\bar{p}_2)$, and we find the non-chiral conformal block

$$\mathcal{G}^{(t)} = \frac{1}{2} \left( \left|\widehat{\mathcal{F}}^{(t)}\right|^2 + \left|\widehat{\mathcal{F}}^{(t)}\right|^2_{(p_2,\bar{p}_2) \to (-p_2,-\bar{p}_2)} \right) \ . \tag{4.34}$$

The violation of chiral factorization in the $t$-channel slightly modifies the crossing symmetry equation (2.20), which now reads

$$\frac{1}{2}\widehat{D}^{(t)} = |16^{p_1 p_2}|^2 \, \widehat{D}^{(s)}_{1+2} = \left|16^{-p_1 p_2}\right|^2 \widehat{D}^{(s)}_{1-2} \ . \tag{4.35}$$

It is straightforward to check that these relations hold, if we compute the four-point structure constants from our two-and three-point structure constants (3.23), (4.19) and (4.27). In particular, we have

$$\frac{1}{2}\widehat{D}^{(t)} = (-)^{n_2 w_1 + n_1 \epsilon_1 + n_2 \epsilon_1} 4^{-p_1^2 - \bar{p}_1^2 - p_2^2 - \bar{p}_2^2} \ . \tag{4.36}$$

This check of crossing symmetry is sensitive to the signs of three-point structure constants. In particular, in our derivation of the three-point structure constant (4.27), we could have chosen the sign prefactor $(-)^{n\epsilon_1}$ rather than $(-)^{n\epsilon_2}$. This would have been consistent with crossing symmetry of twist four-point functions, but not of mixed four-point functions.

**T-duality**

The compactified free boson theory is invariant under inverting the radius of compactification $R \to \frac{1}{R}$, as a consequence of the chiral $\mathbb{Z}_2$ symmetry $(J, \bar{J}) \to (-J, \bar{J})$. This invariance is called T-duality in the context of string theory. From the values of the momentums (3.9), we see that T-duality exchanges the vertex sector indices $n, w$.

In the orbifold theory, the four-point function (4.25) is not invariant under $R \to \frac{1}{R}$, and it may appear that T-duality is broken. In fact, while individual four-point functions of twist fields are not invariant, the space of four-point functions is invariant. T-duality is not broken, but it comes with a change of bases in the two-dimensional space of twist fields $(T^0, T^1) \to (\frac{T^0+T^1}{\sqrt{2}}, \frac{T^0-T^1}{\sqrt{2}})$, in addition to the exchange $(n, w) \to (w, n)$.

## 4.4 Correlation functions of Virasoro primary fields

There are two types of Virasoro primary fields: degenerate fields, and higher twist fields. Correlation functions that do not involve higher twist fields are obtained from the compactified free boson theory using the recipe (4.18). We therefore focus on higher twist fields.

**Two- and three-point structure constants**

The number of higher twist fields in a non-vanishing correlation function is always even: let us therefore consider three-point functions with two higher twist fields. The third field is either a vertex operator, or a degenerate field from the identity sector. Let us write the corresponding three-point structure constants as

$$\boxed{\left\langle T^\epsilon_{r,\bar{r}} T^{\epsilon'}_{r',\bar{r}'} V_{(n,w)} \right\rangle = C_{\epsilon\epsilon'(n,w)} (-)^{r^2-\bar{r}^2} |f_{r,r',p}|^2} \,, \tag{4.37}$$

$$\boxed{\left\langle T^\epsilon_{r,\bar{r}} T^\epsilon_{r',\bar{r}'} I_{k,\bar{k}} \right\rangle = (-)^{r^2-\bar{r}^2} |h_{r,r',k}|^2} \,, \tag{4.38}$$

where $C_{\epsilon\epsilon'(n,w)}$ (4.27) is an affine primary structure constant, $p, \bar{p}$ are the left and right momentums of $V_{(n,w)}$, and $f_{r,r',p}, h_{r,r',k}$ are chiral structure constant.

After factoring out the sign $(-)^{r^2-\bar{r}^2}$, we will find that the remaining factor $|f_{r,r',p}|^2$ is chirally factorized, behaves trivially under permutations i.e. $f_{r,r',p} = f_{r',r,p}$, and obeys $f_{r,r',0} = \delta_{r,r'}$. As a result, the two-point structure constant for higher twist fields is

$$B_{T^\epsilon_{r,\bar{r}}} = (-)^{r^2-\bar{r}^2} \,. \tag{4.39}$$

The violation of chiral factorization by the sign $(-)^{r^2-\bar{r}^2}$ can be traced back to sign ambiguities due to half-integer current modes such as $J_{-\frac{1}{2}}$. These ambiguities affect the correlation functions of twist fields with half-integer spins such as $T^\epsilon_{\frac{3}{4},\frac{1}{4}}$ (4.5), which do not belong to the orbifold CFT. As a result, we cannot factorize correlation functions of $T^\epsilon_{\frac{3}{4},\frac{3}{4}} = 4J_{-\frac{1}{2}} \bar{J}_{-\frac{1}{2}} T^\epsilon_{\frac{1}{4},\frac{1}{4}}$ into correlation functions of $T^\epsilon_{\frac{3}{4},\frac{1}{4}}$ and $T^\epsilon_{\frac{1}{4},\frac{3}{4}}$.

We can compute a few structure constants directly, using the affine Ward identities:

$$h_{\frac{1}{4},\frac{3}{4},1} = \frac{1}{\sqrt{2}} \ , \tag{4.40}$$

$$h_{\frac{1}{4},\frac{5}{4},1} = \frac{1}{\sqrt{2}} \ , \tag{4.41}$$

$$h_{\frac{1}{4},\frac{1}{4},2} = -\frac{1}{2^7\sqrt{6}} \ , \tag{4.42}$$

$$h_{\frac{1}{4},\frac{3}{4},3} = -\frac{3}{2^{16}\sqrt{5}} \ . \tag{4.43}$$

**Chiral crossing symmetry with non-degenerate fields**

To determine the structure constants (4.37), let us allow higher twist fields in the mixed four-point function (4.30), which becomes $\left\langle V_{(n_1,w_1)}V_{(n_2,w_2)}T^{\epsilon_2}_{r_2,\bar{r}_2}T^{\epsilon_1}_{r_1,\bar{r}_1}\right\rangle$. The $s$- and $t$-channel decompositions now look as follows:

$$\tag{4.44}$$

For this four-point function, the crossing symmetry equation (2.20) reads

$$\sum_{\pm} \widehat{D}^{(s)}_{1\pm2}\,|f_{r_1,r_2,p_1\pm p_2}|^2\,|F_{p_1\pm p_2,r}|^2 = \begin{cases} \widehat{D}^{(t)}|f_{r_1,r,p_1}|^2|f_{r_2,r,p_2}|^2 & \text{if } r^2 - \bar{r}^2 \in \mathbb{Z} \ , \\ 0 & \text{if } r^2 - \bar{r}^2 \in \mathbb{Z} + \frac{1}{2} \ , \end{cases} \tag{4.45}$$

where the sign factors from the two-and three-point structure constants cancel, and we use the abbreviated notation $F_{p,r} = F_{p,r}\begin{bmatrix} p_2 & r_2 \\ p_1 & r_1 \end{bmatrix}$ for the fusion kernel. We can factor out the affine four-point structure constants $\widehat{D}^{(s)}_{1\pm2}, \widehat{D}^{(t)}$ using the relation (4.35). We would then like to write a chiral relation between the fusion kernel and the chiral structure constants, rather than a relation involving products of left-moving and right-moving quantities. There is however an ambiguity due to the constraint $r^2 - \bar{r}^2 \in \mathbb{Z}$, which leaves us free to include sign factors of the type $(-)^{2r^2 - \frac{1}{8}}$ in the chiral relation. For simplicity of later expressions, we include such a factor, and write

$$f_{r_1,r_2,p_1+p_2}16^{-p_1p_2}F_{p_1+p_2,r} = f_{r_1,r_2,p_1-p_2}16^{p_1p_2}F_{p_1-p_2,r} = (-)^{2r^2-\frac{1}{8}}f_{r_1,r,p_1}f_{r_2,r,p_2} \ . \tag{4.46}$$

Since no degenerate fields appear as $s$- or $t$-channel fields, the fusion kernel can be determined as a limit of the general fusion kernel, and is given by Eq. (B.24). Using the $G$-function identity (2.30), the solution for the chiral structure constants is

$$\boxed{f_{r_1,r_2,p} = \prod_{i=1}^{2} \frac{\pi^{r_i-\frac{1}{4}}G(\frac{3}{2})}{G(1+2r_i)} \times \prod_{\pm,\pm} \frac{G(1+p \pm r_1 \pm r_2)}{G(1+p \pm \frac{1}{4} \pm \frac{1}{4})}} \ , \tag{4.47}$$

or equivalently,

$$f_{r_1,r_2,p} = \prod_{i=1}^{2} \frac{\pi^{r_i-\frac{1}{4}}G(\frac{3}{2})}{G(1+2r_i)} \times \prod_{\pm} \prod_{r=1-|r_1\pm r_2|}^{|r_1\pm r_2|-1} (p+r)^{|r_1\pm r_2|-|r|} \ , \tag{4.48}$$

where the product over $r \in \frac{1}{2}\mathbb{Z}$ runs by increments of 1. This expression obeys the following properties:

$$f_{\frac{1}{4},\frac{1}{4},p} = 1 \ , \tag{4.49}$$

$$f_{\frac{1}{4},\frac{3}{4},p} = 2p \ , \tag{4.50}$$

$$f_{\frac{1}{4},\frac{5}{4},p} = \frac{2}{3}p(4p^2 - 1) \ , \tag{4.51}$$

$$f_{r_1,r_2,p} = f_{r_2,r_1,p} \ , \tag{4.52}$$

$$f_{r_1,r_2,0} = \delta_{r_1,r_2} \ , \tag{4.53}$$

$$f_{r_1,r_2,-p} = (-)^{2r_1^2 - 2r_2^2} f_{r_1,r_2,p} \ . \tag{4.54}$$

We insist that $f_{r_1,r_2,p}$ is defined up to sign factors of the type $(-)^{2r_i^2 - \frac{1}{8}}$, because the combination that appears in correlation functions is $|f_{r_1,r_2,p}|^2$ with $r_i^2 - \bar{r}_i^2 \in \mathbb{Z}$.

**Chiral crossing symmetry with degenerate fields**

Let us solve crossing symmetry equations for four-point functions $\langle I_1 I_k T_{r_1} T_{r_2} \rangle$,

$$\sum_{\epsilon_1 = \pm} g_{1,k,k+\epsilon_1} h_{r_1,r_2,k+\epsilon_1} F_{\epsilon_1,\epsilon} \begin{bmatrix} k & r_1 \\ 1 & r_2 \end{bmatrix} = \begin{cases} h_{r_2,r_2+\epsilon,1} h_{r_1,r_2+\epsilon,k} & \text{if } \epsilon \in \{+,-\} \ , \\ 0 & \text{if } \epsilon = 0 \ . \end{cases} \tag{4.55}$$

The relevant fusion kernels are given in Eq. (5.47). We use the second equation to get the $k$-dependence of $h_{r_1,r_2,k}$, and guess an ansatz that we check using the first equation. We find the solution

$$\boxed{h_{r_1,r_2,k} = 2^k (-)^{r_1^2 + r_2^2 - \frac{1}{8} - \frac{k}{2}} \pi^{r_1 + r_2 - k - 1} \frac{\prod_{\pm,\pm} G(1 + k \pm r_1 \pm r_2)}{G(1+2k) \prod_{i=1}^{2} G(1+2r_i)} \frac{\prod_{\pm} \Gamma(\frac{k+1 \pm (r_1 \pm_\mathbb{Z} r_2)}{2})}{\sqrt{\Gamma(1+2k)}} \ ,}$$
$$\tag{4.56}$$

where the notation $r_1 \pm_\mathbb{Z} r_2$ was defined in Eq. (4.17). Special cases include

$$h_{r,r,0} = 1 \ , \tag{4.57}$$

$$h_{r,r+\epsilon,1} = \frac{(-)^{2r+\frac{1}{2}}}{\sqrt{2}} \qquad (\epsilon = \pm) \ , \tag{4.58}$$

$$h_{\frac{1}{4},\frac{1}{4},k} = (-)^{\frac{k}{2}} 4^{k-k^2} \frac{\Gamma(\frac{k+1}{2})^2}{\pi \sqrt{\Gamma(2k+1)}} \qquad (\text{for } k \in 2\mathbb{N}) \ , \tag{4.59}$$

$$h_{\frac{1}{4},\frac{1}{4},2} = -\frac{1}{2^7 \sqrt{6}} \ . \tag{4.60}$$

In particular, we find agreement with the value (4.40)-(4.43) that were directly computed using affine Ward identities. Moreover, we have

$$k < |r_1 \pm_\mathbb{Z} r_2| \implies h_{r_1,r_2,k} = 0 \ , \tag{4.61}$$

i.e. the chiral structure constant $h_{r_1,r_2,k}$ vanishes whenever the fusion rule (4.16) is violated. In general, structure constants do not necessarily have to know the fusion rules. In generalized minimal models, some structure constants do, and some of them don't [10, 9]. In our case, the chiral structure constant $h_{r_1,r_2,k}$ determines the contribution of the degenerate field to an OPE of two twist fields, when we compute this contribution using affine symmetry. Affine symmetry dictates that the field $I_k$ is degenerate (i.e. its null vector vanishes), so affine symmetry has to enforce the resulting fusion rules, and this is what we find here.

## 4.5 Examples of correlation functions of higher twist fields

From our structure constants, and the conformal blocks of Appendix A.1, let us assemble a few four-point functions of higher twist fields (4.8). These four-point functions turn out to be related to four-point functions of the affine primary twist fields $T^\epsilon = T^\epsilon_{\frac{1}{4},\frac{1}{4}}$, via relations that involve derivatives with respect to the radius $R$ and the elliptic nome $q$. These relations make crossing symmetry of the new four-point functions manifest.

**Examples of the type $\left\langle T_{\frac{3}{4},\frac{3}{4}} TTT \right\rangle$**

From the structure constants (4.50) and the conformal blocks (A.28), we assemble the four-point functions

$$\left\langle T^{\epsilon_1}_{\frac{3}{4},\frac{3}{4}} \prod_{i=2}^4 T^{\epsilon_i} \right\rangle =$$
$$\delta_{\sum \epsilon_i, 0} \left| \frac{F_0(q)}{x^{1/2}(1-x)^{1/2}\vartheta_3(\tau)^2} \right|^2 \sum_{\substack{n \in \mathbb{Z} \\ w \in 2\mathbb{Z}+\epsilon_1+\epsilon_2}} (-)^{n(\epsilon_2+\epsilon_3)} \left| q^{p_{(n,w)}} \right|^2 \left| 2p_{(n,w)} \right|^2 \quad . \quad (4.62)$$

We notice that the summand can be rewritten as an $R$-derivative, thanks to the identity

$$\left| q^{p_{(n,w)}} \right|^2 \left| 2p_{(n,w)} \right|^2 = -\frac{1}{\pi \operatorname{Im} \tau} R \frac{\partial}{\partial R} \left| q^{p_{(n,w)}} \right|^2 \quad , \quad (4.63)$$

which follows from

$$R \frac{\partial}{\partial R} \, p^2_{(n,w)} = R \frac{\partial}{\partial R} \, \bar{p}^2_{(n,w)} = 2 \left| p_{(n,w)} \right|^2 \quad (4.64)$$

This allows us to relate our four-point function to $\left\langle \prod_{i=1}^4 T^{\epsilon_i} \right\rangle$ (4.25),

$$\boxed{\left\langle T^{\epsilon_1}_{\frac{3}{4},\frac{3}{4}} \prod_{i=2}^4 T^{\epsilon_i} \right\rangle = - \left| \frac{1}{x^{1/2}(1-x)^{1/2}\vartheta_3(\tau)^2} \right|^2 \frac{1}{\operatorname{Im} \tau} R \frac{\partial}{\partial R} \left\langle \prod_{i=1}^4 T^{\epsilon_i} \right\rangle} \quad . \quad (4.65)$$

Since $\left\langle \prod_{i=1}^4 T^{\epsilon_i} \right\rangle$ is known to be crossing-symmetric, the crossing symmetry of our four-point function reduces to the invariance of $\left| \vartheta_3(\tau) \right|^4 \operatorname{Im} \tau$ under the transformation $x \to 1-x$ i.e. $\tau \to -\frac{1}{\tau}$, which we denote $S$. This invariance follows from the identities

$$S \circ \vartheta_3(\tau) = \sqrt{-i\tau}\,\vartheta_3(\tau) \quad , \quad S \circ \operatorname{Im}(\tau) = \frac{1}{|\tau|^2} \operatorname{Im} \tau \quad . \quad (4.66)$$

**Examples of the type $\left\langle T_{\frac{5}{4},\frac{3}{4}} TTT \right\rangle$ and $\left\langle T_{\frac{5}{4},\frac{5}{4}} TTT \right\rangle$**

From the structure constants (4.50), (4.51) and the conformal blocks (A.28), (A.29), we assemble the four-point function

$$\left\langle T^{\epsilon_1}_{\frac{5}{4},\frac{3}{4}} \prod_{i=2}^4 T^{\epsilon_i} \right\rangle = \frac{4}{3} \frac{1}{x(1-x)\vartheta_3(\tau)^4} \left| \frac{F_0(q)}{x^{1/2}(1-x)^{1/2}\vartheta_3(\tau)^2} \right|^2 \times$$
$$\delta_{\sum \epsilon_i, 0} \sum_{\substack{n \in \mathbb{Z} \\ w \in 2\mathbb{Z}+\epsilon_1+\epsilon_2}} (-)^{n(\epsilon_2+\epsilon_3)} \left| q^{p_{(n,w)}} \right|^2 \left| 2p_{(n,w)} \right|^2 \left( p^2_{(n,w)} - q \frac{\partial}{\partial q} \log \vartheta'_1(\tau) \right) \quad . \quad (4.67)$$

This expression can again be represented as a differential operator acting on $\left\langle \prod_{i=1}^{4} T^{\epsilon_i} \right\rangle$. To do this, we combine the $R$-derivative identity (4.63) with the identity

$$q^{p^2}\left(p^2 - q\frac{\partial}{\partial q}\log \vartheta_1'(\tau)\right) = \vartheta_1'(\tau)q\frac{\partial}{\partial q}\frac{q^{p^2}}{\vartheta_1'(\tau)} \;, \tag{4.68}$$

and we obtain the relation

$$\left\langle T_{\frac{5}{4},\frac{3}{4}}^{\epsilon_1}\prod_{i=2}^{4} T^{\epsilon_i} \right\rangle = -\frac{4}{3\pi}\left|\frac{1}{x^{1/2}(1-x)^{1/2}\vartheta_3(\tau)^2}\right|^2 \times$$
$$\frac{\vartheta_1'(\tau)F_0(q)}{x(1-x)\vartheta_3(\tau)^4}\left(q\frac{\partial}{\partial q}\right)\frac{1}{\vartheta_1'(\tau)F_0(q)\operatorname{Im}\tau}\left(R\frac{\partial}{\partial R}\right)\left\langle \prod_{i=1}^{4} T^{\epsilon_i} \right\rangle \;. \tag{4.69}$$

In addition to an $R$-derivative, this relation also involves a $q$-derivative. In order to check that it is crossing-symmetric, we need not only the identities (4.66), but also the following identities:

$$S \circ \vartheta_1'(\tau) = i(-i\tau)^{3/2}\vartheta_1'(\tau) \quad , \quad S \circ F_0(q) = \frac{1}{\sqrt{-i\tau}}F_0(q) \quad , \quad S \circ q\frac{\partial}{\partial q} = \tau^2 q\frac{\partial}{\partial q} \;. \tag{4.70}$$

The case of the higher twist field $T_{\frac{5}{4},\frac{5}{4}}^{\epsilon}$ is similar to the case of $T_{\frac{5}{4},\frac{3}{4}}^{\epsilon}$, and is in fact a bit simpler due to its left-right symmetry. In this case, the relation with $\left\langle \prod_{i=1}^{4} T^{\epsilon_i} \right\rangle$ reads

$$\left\langle T_{\frac{5}{4},\frac{5}{4}}^{\epsilon_1}\prod_{i=2}^{4} T^{\epsilon_i} \right\rangle = -\frac{16}{9\pi}\left|\frac{F_0(q)\vartheta_1'(\tau)}{x^{1/2}(1-x)^{1/2}\vartheta_3(\tau)^6}\right|^2 \times$$
$$\left|q\frac{\partial}{\partial q}\right|^2\frac{1}{|F_0(q)\vartheta_1'(\tau)|^2\operatorname{Im}\tau}R\frac{\partial}{\partial R}\left\langle \prod_{i=1}^{4} T^{\epsilon_i} \right\rangle \;. \tag{4.71}$$

As before, crossing symmetry of this expression is readily verified from the transformation properties (4.66), (4.70).

**Interpretation**

Our derivation of four-point functions of higher twist fields was purely technical, but perhaps there is an underlying reason for their simplicity. By definition, higher twist fields are obtained from basic twist fields by acting with modes of the currents $J$ and $\bar{J}$. The $R$-derivatives that appear in our relations can be interpreted in terms of the marginal operator $J\bar{J}$. And the $q$-derivatives can be interpreted in terms of the energy-momentum tensor $JJ$, if we recall the relation of twist correlators with correlators on a torus with modulus $\tau$ [5], together with the torus Ward identities [17]. It would be interesting to further explore the interplay between twist fields, target space geometry, and worldsheet geometry.

# 5    Degenerate Virasoro conformal blocks at $c = 1$

Let us study Virasoro conformal blocks whose channel dimensions are degenerate. If $c = 1$, the momentums of degenerate representations are of the type $k \in \frac{1}{2}\mathbb{Z}$. In the

Ashkin–Teller model and compactified free boson, degenerate representations appear in the identity sector, and we denoted the corresponding fields as $I_{k,\bar{k}}$ with $k, \bar{k} \in \mathbb{N}$.

Our strategy will be to take limits from non-degenerate conformal blocks, which can be computed using general methods such as Zamolodchikov's recursive representation. We will show that appropriate limits exist, although this is far from trivial.

## 5.1 Analytic subtleties of conformal blocks

### Conformal blocks as sums over states

Consider a four-point function of Virasoro primary fields $\left\langle \prod_{i=1}^{4} V_{\Delta_i}(z_i) \right\rangle$ with $(z_i) = (0, z, 1, \infty)$. The corresponding $s$-channel blocks depend on the central charge $c$, on the four fields, and on a Virasoro representation $\mathcal{R}_s$. We assume that $\mathcal{R}_s$ is a highest-weight representation, i.e. it is generated by a primary state of dimension $\Delta_s$. Then the blocks can be written as

$$\mathcal{F}_{\mathcal{R}_s}(c|(\Delta_i)|z) = z^{\Delta_s - \Delta_1 - \Delta_2}$$
$$\times \sum_{Y,Y' \in B(\mathcal{R}_s)} z^{|Y|} \gamma_Y(\Delta_1, \Delta_2 | \Delta_s) G_{Y,Y'}^{-1}(c|\Delta_s) \gamma_{Y'}(\Delta_3, \Delta_4 | \Delta_s) . \quad (5.1)$$

where we introduced the following objects:

- $B(\mathcal{R}_s)$ is a set of creation operators such that $B(\mathcal{R}_s)|\Delta_s\rangle$ is a basis of $\mathcal{R}_s$, if $|\Delta_s\rangle$ is the primary state of dimension $\Delta_s$. We assume that the resulting basis is made of $L_0$-eigenvectors, i.e. $[L_0, Y] = |Y|Y$ where $|Y| \in \mathbb{N}$ is the level, and $\Delta_s + |Y|$ the conformal dimension of $Y|\Delta_s\rangle$.

- $\gamma_Y(\Delta_1, \Delta_2 | \Delta_3)$ is a $c$-independent polynomial function of three conformal dimensions.

- $G_{Y,Y'}^{-1}(c|\Delta_s)$ is the inverse Gram matrix. The Gram matrix itself depends polynomially on $c$ and $\Delta_s$, and its inverse has poles.

In the special case where $\mathcal{R}_s = \mathcal{V}_{\Delta_s}$ is a Verma module, we have

$$B(\mathcal{V}_{\Delta_s}) = 1, L_{-1}, L_{-1}^2, L_{-2}, L_{-1}^3, L_{-1}L_{-2}, L_{-3}, \dots \quad (5.2)$$

In this case, the object $\mathcal{F}_{\Delta_s} \equiv \mathcal{F}_{\mathcal{V}_{\Delta_s}}$ is called the Virasoro conformal block for the channel dimension $\Delta_s$: we will call it a generic block. This universal function can be computed using a number of techniques, starting with Zamolodchikov's recursive representation. (Directly using Eq. (5.1) is not efficient.)

Our aim is to compute conformal blocks for degenerate representations, by relating them to the well-known generic blocks $\mathcal{F}_{\Delta_s}$. A degenerate representation is a highest-weight representations, i.e. a quotient of a Verma module by a submodule. So the basis $B(\mathcal{R}_s)$ is a subset of $B(\mathcal{V}_{\Delta_s})$. The other ingredients of the expression (5.1) only depend on $\Delta_s$, so they are the same for $\mathcal{R}_s$ as for $\mathcal{V}_{\Delta_s}$. Beware however that $G_{Y,Y'}^{-1}(c|\Delta_s)$ is the inverse of a submatrix of the Gram matrix, which is not the same as a submatrix of the inverse Gram matrix. (The Gram matrix of a degenerate representation is not invertible.)

### Singularities of the inverse Gram matrix at level 2

In order to illustrate the analytic properties of conformal blocks, let us focus on the contributions from states at level 2. This is the first nontrivial case, as the Gram matrix

at level 1 happens to be $c$-independent. Let us introduce notations for the central charge and conformal dimensions, which will also be convenient in the rest of Section 5. We write the central charge as

$$c = 13 - 6\beta^2 - 6\beta^{-2} \ . \tag{5.3}$$

The conformal dimension is written in terms of a momentum $p$, via a relation which generalizes the $c = 1$ relation (2.21),

$$\Delta = \frac{c-1}{24} + p^2 \ . \tag{5.4}$$

The degenerate momentums are of the type

$$p_{(r,s)} = \frac{1}{2} \left( r\beta - s\beta^{-1} \right) \ . \tag{5.5}$$

For $r, s \in \mathbb{N}^*$, these are the momentums of degenerate representations, and they become half-integer as $c \to 1$ i.e. $\beta \to 1$.

For $\Delta = \Delta_{(2,1)} = -\frac{1}{2} + \frac{3}{4}\beta^2$, there exists a degenerate representation $\mathcal{R}_{\langle 2,1 \rangle} = \frac{\mathcal{V}_{\Delta_{(2,1)}}}{\mathcal{V}_{\Delta_{(2,1)}+2}}$, which is the quotient of the Verma module by the submodule generated by the level 2 null vector $\chi_{\langle 2,1 \rangle} = Y_{\langle 2,1 \rangle} |\Delta_{(2,1)}\rangle$, where $Y_{\langle 2,1 \rangle} = L_{-2} - \beta^{-2} L_{-1}^2$. In order to study the limit $\Delta \to \Delta_{\langle 2,1 \rangle}$, we therefore write the level 2 Gram matrix in the basis $B^{(2)}(\mathcal{V}_\Delta) = (L_{-2}, Y_{\langle 2,1 \rangle})$, and rewrite $\Delta = \Delta_{\langle 2,1 \rangle} + \epsilon$. We find

$$G^{(2)} = \begin{bmatrix} G_{L_{-2},L_{-2}} & G_{L_{-2},Y_{\langle 2,1 \rangle}} \\ G_{Y_{\langle 2,1 \rangle},L_{-2}} & G_{Y_{\langle 2,1 \rangle},Y_{\langle 2,1 \rangle}} \end{bmatrix} = \begin{bmatrix} \frac{9}{2} - 3\beta^{-2} + 4\epsilon & (2 - 3\beta^{-2})\epsilon \\ (2 - 3\beta^{-2})\epsilon & 4(1 - \beta^{-4})\epsilon + 8\beta^{-4}\epsilon^2 \end{bmatrix} \ . \tag{5.6}$$

The behaviour of the inverse Gram matrix crucially depends on whether $c = 1$ or not, because $G_{Y_{\langle 2,1 \rangle},Y_{\langle 2,1 \rangle}}$ behaves as $O(\epsilon)$ in general, and $O(\epsilon^2)$ if $c = 1$. As a result, we have

$$\left( G^{(2)} \right)^{-1} \underset{c \neq 1}{=} \begin{bmatrix} \frac{1}{\frac{9}{2} - 3\beta^{-2}} + O(\epsilon) & O(1) \\ O(1) & \frac{1}{4(1 - \beta^{-4})\epsilon} + O(1) \end{bmatrix} \ , \tag{5.7}$$

so that in this case

$$\lim_{\epsilon \to 0} \left( G^{(2)} \right)^{-1}_{L_{-2},L_{-2}} \underset{c \neq 1}{=} \frac{1}{\lim_{\epsilon \to 0} G^{(2)}_{L_{-2},L_{-2}}} \ . \tag{5.8}$$

This allows us to compute the inverse of the size one submatrix $\left[ \lim_{\epsilon \to 0} G^{(2)}_{L_{-2},L_{-2}} \right]$, which appears in the degenerate block $\mathcal{F}_{\mathcal{R}_{\langle 2,1 \rangle}}$, from the inverse matrix $\left( G^{(2)} \right)^{-1}$, which appears in the generic block $\mathcal{F}_\Delta$. This no longer works if $c = 1$, in which case

$$G^{(2)} \underset{c=1}{=} \begin{bmatrix} \frac{3}{2} + 4\epsilon & -\epsilon \\ -\epsilon & 8\epsilon^2 \end{bmatrix} \quad , \quad \left( G^{(2)} \right)^{-1} \underset{c=1}{=} \begin{bmatrix} \frac{8}{11} + O(\epsilon) & O\left(\frac{1}{\epsilon}\right) \\ O\left(\frac{1}{\epsilon}\right) & O\left(\frac{1}{\epsilon^2}\right) \end{bmatrix} \ , \tag{5.9}$$

so that

$$\lim_{\epsilon \to 0} \left( G^{(2)} \right)^{-1}_{L_{-2},L_{-2}} \underset{c=1}{=} \frac{8}{11} \neq \frac{2}{3} \underset{c=1}{=} \frac{1}{\lim_{\epsilon \to 0} G^{(2)}_{L_{-2},L_{-2}}} \ . \tag{5.10}$$

Ultimately, the reason why the level two Gram matrix behaves badly if $c = 1$ is the coincidence of the degenerate dimensions for which null vectors appear at this level, $\Delta_{(2,1)} \underset{c=1}{=} \Delta_{(1,2)}$.

**Degenerate blocks from generic blocks: two conjectures**

Let us now discuss the polynomials $\gamma_Y(\Delta_1, \Delta_2|\Delta_3)$ that appear in the conformal blocks (5.1). By definition of fusion rules, the polynomial $\gamma_{Y_{\langle 2,1 \rangle}}(\Delta_1, \Delta_2|\Delta_s)$ has a zero at $\Delta_s = \Delta_{(2,1)}$ whenever $\mathcal{V}_{\Delta_2} \in \mathcal{R}_{\langle 2,1 \rangle} \times \mathcal{V}_{\Delta_1}$. (This condition amounts to a second-order polynomial equation on $\Delta_1, \Delta_2$ [9].) Of course, we assume that the degenerate blocks we want to compute are allowed by fusion, which implies

$$\gamma_{Y_{\langle 2,1 \rangle}}(\Delta_1, \Delta_2|\Delta_{(2,1)}) = \gamma_{Y_{\langle 2,1 \rangle}}(\Delta_3, \Delta_4|\Delta_{(2,1)}) = 0 . \tag{5.11}$$

If $c \neq 1$, when computing the level two contribution to the generic block $\mathcal{F}_{\Delta_s}(c|(\Delta_i)|z)$ (5.1) in the limit $\Delta_s \to \Delta_{(2,1)}$, the zeros from the $\gamma_{Y_{\langle 2,1 \rangle}}$ factors cancel the contributions of $\left(G^{(2)}\right)^{-1}_{L_{-2}, Y_{\langle 2,1 \rangle}}, \left(G^{(2)}\right)^{-1}_{Y_{\langle 2,1 \rangle}, L_{-2}}, \left(G^{(2)}\right)^{-1}_{Y_{\langle 2,1 \rangle}, Y_{\langle 2,1 \rangle}}$ (5.7). We are left with the contribution of $\left(G^{(2)}\right)^{-1}_{L_{-2}, L_{-2}}$, which agrees with the level two term of the degenerate block $\mathcal{F}_{\mathcal{R}_{\langle 2,1 \rangle}}$ by Eq. (5.8). We conjecture that this is true not only at level two, but at all levels; not only for the degenerate representation $\mathcal{R}_{\langle 2,1 \rangle}$, but for all degenerate representations:

**Conjecture 1**
*If the central charge is irrational, and if the degenerate representation $\mathcal{R}_{\langle r,s \rangle}$ is allowed by fusion rules in the s-channel of $\left\langle \prod_{i=1}^4 V_{\Delta_i}(z_i) \right\rangle$, then the corresponding degenerate block is a limit of generic blocks,*

$$\boxed{\mathcal{F}_{\mathcal{R}_{\langle r,s \rangle}}(c|(\Delta_i)|z) = \lim_{\Delta \to \Delta_{(r,s)}} \mathcal{F}_\Delta(c|(\Delta_i)|z)} . \tag{5.12}$$

We know that the conjecture cannot hold if the central charge is rational, due to coincidences of degenerate dimensions $\Delta_{(r,s)} = \Delta_{(r',s')}$. (To be precise, by rational central charge we mean $\beta^2 \in \mathbb{Q}$.) To compute degenerate blocks at rational central charge, we propose to first continue them to generic central charge, then take the limit $\lim_{\Delta \to \Delta_{(r,s)}}$, and then go back to rational central charge:

**Conjecture 2**
*Let $c_0$ be a rational central charge, and $\mathcal{R}$ a degenerate representation that is allowed in the s-channel of $\left\langle \prod_{i=1}^4 V_{\Delta_i}(z_i) \right\rangle$. Let $\mathcal{R}^{(c)}$ be any degenerate representation at generic $c$ that has the same number of states at each level as $\mathcal{R}$. Let $(\Delta_i^{(c)})$ be a continuation of $(\Delta_i)$ such that $\mathcal{R}^{(c)}$ is allowed in the s-channel of $\left\langle \prod_{i=1}^4 V_{\Delta_i^{(c)}}(z_i) \right\rangle$. Then we have*

$$\boxed{\mathcal{F}_{\mathcal{R}}(c_0|(\Delta_i)|z) = \lim_{c \to c_0} \mathcal{F}_{\mathcal{R}^{(c)}}(c|(\Delta_i^{(c)})|z)} . \tag{5.13}$$

Let us motivate this conjecture, and in particular the assumption that $\mathcal{R}^{(c)}$ has the same number of states at each level as $\mathcal{R}$. To obtain the desired degenerate block $\mathcal{F}_{\mathcal{R}}(c_0|(\Delta_i)|z)$, we need to reduce the basis $B(\mathcal{V}_\Delta)$ to the smaller basis $B(\mathcal{R})$, i.e. we need the terms from $B(\mathcal{V}_\Delta) - B(\mathcal{R})$ to vanish. For this we must do an excursion to irrational central charge.

We may worry that this excursion may deform $B(\mathcal{R})$. However, this does not matter, because $\mathcal{R}$ is a quotient space, whose elements are only defined modulo null vectors. In our level two example, we chose $L_{-2}$ as the basis of $B^{(2)}(\mathcal{R}_{\langle 2,1 \rangle})$, but we could have chosen any $c$-dependent vector that is not collinear to $Y_{\langle 2,1 \rangle}$. To get a basis of $\mathcal{R}$, it only matters that we have the right numbers of states at each level.

This justifies our assumption on $\mathcal{R}^{(c)}$. In order to exploit Conjecture 1, we further assume that fusion rules are obeyed at generic $c$, and this constrains the continued dimensions $(\Delta_i^{(c)})$.

**Scope and validity of the conjectures**

Conjecture 1 is only the explict formulation of a well-known fact. From Zamolodchikov's recursive representation, it is easy to deduce that the limit exists, because the fusion rules ensure $\mathrm{Res}_{\Delta=\Delta_{(r,s)}} \mathcal{F}_\Delta(c|(\Delta_i)|z) = 0$. Conjecture 1 has long been used for numerically computing conformal blocks and testing crossing symmetry in models such as Generalized Minimal Models [7] or the Potts model [18], so there is little doubt that it is true.

Conjecture 2 is newer and less trivial. In this case, the very existence of the limit is already a non-trivial statement. Moreover, Conjecture 2 only provides a particular prescription for computing degenerate blocks from generic blocks. We do not claim that this prescription is the only possibility. And it had better not be: given the intricate structures of degenerate representations at rational $c$, it is in general not possible to find generic $c$ representations $\mathcal{R}^{(c)}$ that satisfy our assumptions. Conjecture 2 is however good enough for the Ashkin–Teller model, whose degenerate representations have relatively simple structures: all null vectors descend from only one singular vector.

We however warn that it is not easy to find alternative prescriptions. For example, we may be tempted to replace (5.13) with the simpler prescription of keeping all conformal dimensions fixed i.e. $c$-independent,

$$\mathcal{F}_{\mathcal{R}}(c_0|(\Delta_i)|z) \overset{?}{=} \lim_{c\to c_0} \mathcal{F}_{\Delta_{\mathcal{R}}}(c|(\Delta_i)|z) \ . \tag{5.14}$$

We have checked that this prescription fails in examples, by giving a finite but wrong limit. However, depending on the case, it may fail at rather high level.

Let us summarize our evidence for Conjecture 2. All our evidence is at $c = 1$. First of all, in Section 5.2, we will apply the conjecture to the computation of conformal blocks of four fields of dimension $\Delta = \frac{1}{16}$ i.e. momentum $p = \frac{1}{4}$, and obtain consistent results which can be compared to computations from affine symmetry to some extent. Then, in Section 5.3, we will use Conjecture 2 for computing degenerate fusion kernels at $c = 1$. These fusion kernels are then used for writing crossing symmetry equations in the Ashkin–Teller model. Our formulas for the structure constants of that model therefore crucially depend on our conjectures. In a number of examples, we can independently derive the same structure constants using the model's affine symmetry. All this provides evidence for the conjecture in the cases of the following families of $c = 1$ Virasoro blocks:

$$\mathcal{F}^{(s)}_{2k|\frac{1}{4},\frac{1}{4},\frac{1}{4},\frac{1}{4}} \ , \quad \mathcal{F}^{(s)}_{k+\epsilon|1,k,r_1,r_2} \ , \quad \mathcal{F}^{(s)}_{k+\epsilon|1,k,p,p} \ , \quad \mathcal{F}^{(s)}_{k_1+\epsilon|1,k_1,k_2,k_3} \ , \tag{5.15}$$

where we label blocks by their momentums, with $k, k_i \in \mathbb{N}$ and $\epsilon \in \{-1, 0, 1\}$ and $r_i \in \frac{1}{4} + \frac{1}{2}\mathbb{Z}$ and $p \in \mathbb{C}$. This evidence is admittedly indirect, but it tests the conjecture at all levels, which would be impossible with pedestrian calculations.

## 5.2 Four-point functions of twist fields

**Virasoro blocks versus affine blocks**

Let us go back to $c = 1$, and consider four-point functions of twist fields of the type $\langle TTTT \rangle$, i.e. four-point functions of fields of dimension $\frac{1}{16}$. The corresponding affine conformal blocks were determined by Al. Zamolodchikov [2], see Eq. (A.26). Such blocks have an extremely simple dependence on the momentum $p$, namely $\widehat{\mathcal{F}}_p \propto (16q)^{p^2}$. For generic values of $p$, the Virasoro and affine representations coincide, and Al. Zamolodchikov's blocks coincide with Virasoro conformal blocks, $\widehat{\mathcal{F}}_p = \mathcal{F}_p$. The limit $p \to 0$ of the blocks is perfectly smooth, and yields the affine conformal block $\widehat{\mathcal{F}}_0$.

From the point of view of the Virasoro algebra, the situation is more complicated. The dimension $p = p_{(1,1)} = 0$ corresponds to the identity field, which is allowed by its fusion rules to appear in our four-point function. If the central charge was generic, $\widehat{\mathcal{F}}_0$ would have to coincide with the identity block $\mathcal{F}_0$ per Conjecture 1. However, the fusion rule (4.16) predicts that the affine block is an infinite linear combination of degenerate Virasoro blocks of the type

$$\widehat{\mathcal{F}}_0 = \sum_{k \overset{2}{=} 0}^{\infty} h_{\frac{1}{4}, \frac{1}{4}, k}^2 \mathcal{F}_k \ , \tag{5.16}$$

where $\mathcal{F}_k$ is a degenerate Virasoro conformal block, and $h_{\frac{1}{4}, \frac{1}{4}, k}$ is a chiral structure constant of the Ashkin–Teller model, defined by Eq. (4.38). In particular, we expect that $\widehat{\mathcal{F}}_0$ and $\mathcal{F}_0$ agree up to level 3, but differ at levels 4 and higher.

**Recursive representation**

Let us use Conjecture 2 for computing the degenerate Virasoro block $\mathcal{F}_k$ at $c = 1$. The degenerate representation with momentum $k$ is the quotient of a Verma module with the Verma submodule that is generated by the null vector at level $2k + 1$. At generic central charge, there exist two degenerate representations with the same structure, namely $\mathcal{R}_{\langle 2k+1,1 \rangle}$ and $\mathcal{R}_{\langle 1,2k+1 \rangle}$. According to the conjecture, it should not matter which degenerate representation we choose, and we pick the second one. We should also continue the twist fields of dimension $\frac{1}{4}$ in a way that respects fusion rules at generic central charge. To satisfy this requirement, it is enough that the momentums of the four twist fields remain equal. For the simplicity of our calculations, we will choose their momentums to be $p_{(0,\frac{1}{2})}$. This leads to

$$\mathcal{F}_k \left( 1 \big| (\tfrac{1}{4}, \tfrac{1}{4}, \tfrac{1}{4}, \tfrac{1}{4}) \big| z \right) = \lim_{c \to 1} \mathcal{F}_{p_{(1,2k+1)}} \left( c \big| (p_{(0,\frac{1}{2})}, p_{(0,\frac{1}{2})}, p_{(0,\frac{1}{2})}, p_{(0,\frac{1}{2})}) \big| z \right) \ . \tag{5.17}$$

Zamolodchikov's recursive representation for Virasoro conformal blocks is

$$\mathcal{F}_p \left( c \big| (p_{(0,\frac{1}{2})}, p_{(0,\frac{1}{2})}, p_{(0,\frac{1}{2})}, p_{(0,\frac{1}{2})}) \big| z \right) \propto (16q)^{p^2} H_p(q) \ , \tag{5.18}$$

where we neglect $p$-independent factors, and the power series $H_p(q)$ is defined recursively by

$$H_p(q) = 1 + \sum_{r \overset{2}{=} 0}^{\infty} \sum_{s=0}^{\infty} \frac{R_{r,s}(16q)^{rs}}{p^2 - p_{(r,s)}^2} H_{p_{(r,-s)}}(q) \ . \tag{5.19}$$

With our choice of momentums, the first summation index $r$ only takes even values, and the residues $R_{r,s}$ take the form [19](Appendix A.1)

$$R_{r,s} = -2^{1-4rs} p_{(0,0)} p_{(r,s)} \prod_{r'=1-r}^{r} \prod_{s'=1-s}^{s} p_{(r',s')}^{(-1)^{r'}+1} \ . \tag{5.20}$$

Let us investigate the behaviour of $H_p(q)$ in the limit $c \to 1$. Due to the relation $p_{(1-r',1-s')} = p_{(1,1)} - p_{(r',s')}$ with $\lim p_{(1,1)} = 0$, the factors with $r' \neq s'$ in $R_{r,s}$ cancel, and only the diagonal $r' = s'$ contribute. We find

$$R_{r,s} \underset{c \to 1}{\sim} 2^{1-4rs} p_{(r,s)} p_{(1,1)} \tilde{R}_{\min(r,s)} \qquad \text{with} \qquad \boxed{\tilde{R}_m = \frac{4\Gamma(\frac{m+1}{2})^2}{m\Gamma(\frac{1}{2})^2 \Gamma(\frac{m}{2})^2}} \ . \tag{5.21}$$

Since $\lim_{c \to 1} p_{(r,r)} = 0$, we have $\lim_{c \to 1} R_{r,s} = 0$, with a zero of order one if $r \neq s$ and two if $r = s$. In the recursive representation (5.19) of $H_{p_{(m,n)}}(q)$ with $m, n \in \mathbb{Z}$, we find that the coefficients have the limits

$$\lim_{c \to 1} \frac{R_{r,s} 16^{rs}}{p_{(m,n)}^2 - p_{(r,s)}^2} = - \left( \frac{\delta_{m-n,r-s}}{r - m} + \frac{\delta_{m-n,s-r}}{r + m} \right) \tilde{R}_{\min(r,s)} \ . \tag{5.22}$$

We can therefore define the finite limit

$$H_{m,n}(q) = \lim_{c \to 1} H_{p_{(m,n)}}(q) \qquad \text{if} \qquad (m, n) \in \mathbb{Z}^2 - (2\mathbb{N}^* \times \mathbb{N}^*) \ . \tag{5.23}$$

Moreover, since the recursive representation of $H_{p_{(m,n)}}(q)$ is finite term by term, its limit $H_{m,n}(q)$ also has a recursive representation. We write this representation in the case $m - n \in 2\mathbb{N}$, which is enough for our purposes:

$$H_{m,n}(q) \underset{m-n \in 2\mathbb{N}}{=} 1 - \sum_{r \overset{2}{=} 0}^{\infty} \tilde{R}_r \left( \frac{1}{r + m} + \frac{1}{r - n} \right) q^{r(m-n+r)} H_{m-n+r,-r}(q) \ , \tag{5.24}$$

where we used the identities $H_{m,n}(q) = H_{-m,-n}(q) \underset{m-n \in 2\mathbb{N}}{=} H_{n,m}(q)$, and $\tilde{R}_r$ was defined in Eq. (5.21). This might be the first known instance of a recursive representation for conformal blocks at rational central charge.

**Generalized theta functions**

Using our recursive representation, we can compute the degenerate Virasoro conformal blocks $\mathcal{F}_k \left( 1 \big| (\frac{1}{4}, \frac{1}{4}, \frac{1}{4}, \frac{1}{4}) \big| z \right) \propto (16q)^{k^2} H_{1,2k+1}(q)$ with $k \in 2\mathbb{N}$ to any given order. Examining the results, we infer an expression for these blocks to all orders,

$$q^{k^2} H_{1,2k+1}(q) = - \frac{\Gamma(k + \frac{3}{2})}{2 \Gamma(k+1) \Gamma(\frac{1}{2})} \sum_{m \in \mathbb{Z}} \frac{\Gamma(m - \frac{k+1}{2})}{\Gamma(m + 1 + \frac{k+1}{2})} \frac{m \Gamma(m + \frac{k}{2})}{\Gamma(m + 1 - \frac{k}{2})} q^{4m^2} \ . \tag{5.25}$$

In these expressions, the terms with $|m| < \frac{k}{2}$ vanish, due to the zeros of the polynomial

$$\frac{m \Gamma(m + \frac{k}{2})}{\Gamma(m + 1 - \frac{k}{2})} = \prod_{i=0}^{\frac{k}{2}-1} (m^2 - i^2) \ . \tag{5.26}$$

The first two examples are the four-twist identity block

$$H_{1,1}(q) = \sum_{m \in \mathbb{Z}} \frac{q^{4m^2}}{1 - 4m^2} = 1 - \frac{2}{3} q^4 - \frac{2}{15} q^{16} - \frac{2}{35} q^{36} - \frac{2}{63} q^{64} - \frac{2}{99} q^{100} - \cdots \ , \tag{5.27}$$

and the block for the degenerate field with a vanishing null vector at level 5,

$$q^4 H_{1,5}(q) = -15 \sum_{m=1}^{\infty} \frac{m^2 q^{4m^2}}{(4m^2 - 1)(4m^2 - 9)} \ . \tag{5.28}$$

We call these functions generalized theta functions, because they can be obtained by acting with integro-differential operators on the theta function $\sum_{m \in \mathbb{Z}} q^{4m^2}$.

Finally, we have checked that the infinite linear combination (5.16) of degenerate blocks, with the coefficients $h^2_{\frac{1}{4},\frac{1}{4},k}$ given by Eq. (4.59), does reproduce the affine block $\widehat{\mathcal{F}}_0$. In terms of the factors $H_{m,n}(q)$, this amounts to the identity

$$1 = \sum_{k \stackrel{2}{=} 0}^{\infty} h^2_{\frac{1}{4},\frac{1}{4},k} (16q)^{k^2} H_{1,2k+1}(q) \ , \tag{5.29}$$

which we have checked numerically to high order. So we have reproduced a trivial affine block as a combination of Virasoro structure constants and degenerate blocks, computed using Conjecture 2. This provides evidence in favour of the conjecture.

## 5.3   Degenerate fusion kernels

In order to write crossing symmetry equations that involve a degenerate field with momentum $k = 1$, we need to compute the corresponding fusion kernels. As in the case of $c = 1$ degenerate conformal blocks, we will compute $c = 1$ degenerate fusion kernels by taking limits from generic central charge. So we will first compute the relevant fusion kernels at generic $c$, which do not appear in the literature.

### Pentagon relation

The degenerate field with $k = 1$ has a vanishing null vector at level 3, and its continuation to generic $c$ must therefore be either $V_{\langle 3,1\rangle}$ or $V_{\langle 1,3\rangle}$. We pick $V_{\langle 3,1\rangle}$, whose fusion rules with a primary field of momentum $p$ are

$$V_{\langle 3,1\rangle} \times V_p \sim \sum_{\epsilon \in \{-1,0,1\}} V_{p+\epsilon\beta} \ . \tag{5.30}$$

A four-point function with $V_{\langle 3,1\rangle}$ obeys a third-order BPZ equation, whose fusion kernel is not that simple. To compute it, we will reduce the problem to the simpler case of the second-order BPZ equation for four-point functions with $V_{\langle 2,1\rangle}$, whose fusion rules are

$$V_{\langle 2,1\rangle} \times V_p \sim \sum_{\eta \in \{+,-\}} V_{p+\frac{\eta}{2}\beta} \ . \tag{5.31}$$

This reduction is possible thanks to the fusion rule

$$V_{\langle 2,1\rangle} \times V_{\langle 2,1\rangle} \sim V_{\langle 3,1\rangle} + V_{\langle 1,1\rangle} \ , \tag{5.32}$$

which implies that the corresponding fusion kernels obey the the Pentagon relation [9]:

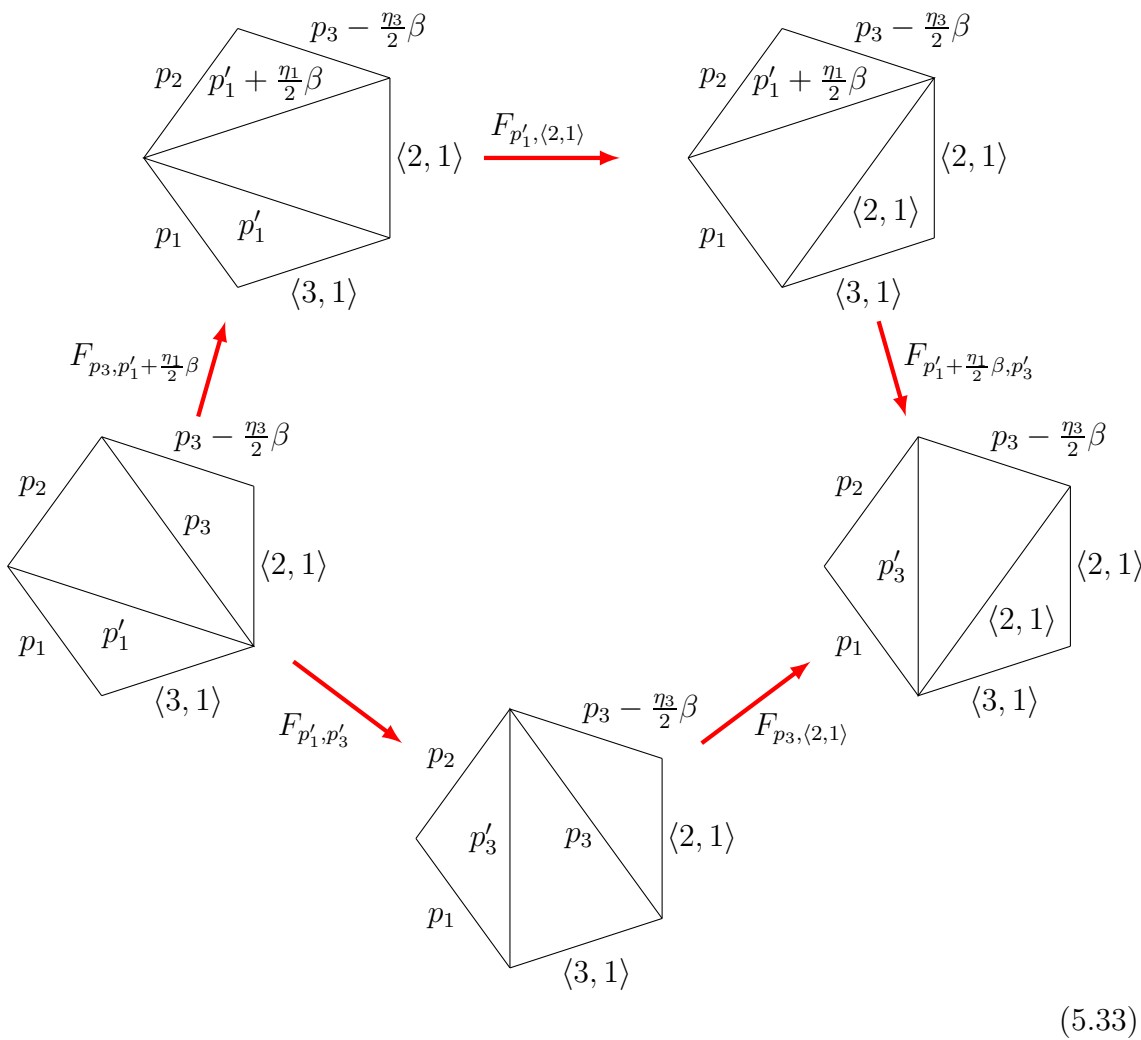

$$(5.33)$$

This diagram depicts two equivalent sequences of fusion moves for five-point conformal blocks. Each black line is a primary field, and each red arrow is a fusion kernel. In this notation, the fusion relation (2.19) would read

$$F_{\Delta_s, \Delta_t} \qquad (5.34)$$

In our Pentagon relation, we have three arbitrary momentums $p_1, p_2, p_3$, two shifted momentums

$$(i = 1, 3) \qquad p_i' = p_i + \epsilon_i \beta \quad \text{with} \quad \epsilon_i \in \{-1, 0, 1\} \,, \tag{5.35}$$

and two signs $\eta_1, \eta_3 \in \{+, -\}$, where the fusion rules of the degenerate fields $V_{\langle 2,1 \rangle}$ and $V_{\langle 3,1 \rangle}$ imply $\epsilon_i \neq 0 \implies \eta_i = -\epsilon_i$. In equation form, our Pentagon relation reads

$$F_{p_1', p_3'} \begin{bmatrix} p_1 & p_2 \\ \langle 3, 1 \rangle & p_3 \end{bmatrix} F_{p_3, \langle 2,1 \rangle} \begin{bmatrix} p_3 - \frac{\eta_3}{2}\beta & p_3' \\ \langle 2, 1 \rangle & \langle 3, 1 \rangle \end{bmatrix} = \sum_{\eta_1 = \pm}$$

$$F_{p_3, p_1' + \frac{\eta_1}{2}\beta} \begin{bmatrix} p_3 - \frac{\eta_3}{2}\beta & p_2 \\ \langle 2, 1 \rangle & p_1' \end{bmatrix} F_{p_1', \langle 2,1 \rangle} \begin{bmatrix} p_1' + \frac{\eta_1}{2}\beta & p_1 \\ \langle 2, 1 \rangle & \langle 3, 1 \rangle \end{bmatrix} F_{p_1' + \frac{\eta_1}{2}\beta, p_3'} \begin{bmatrix} p_1 & p_2 \\ \langle 2, 1 \rangle & p_3 - \frac{\eta_3}{2}\beta \end{bmatrix}.$$

$$(5.36)$$

In this relation, the first fusion kernel is the most general fusion kernel for a four-point function of the type $\langle V_{\langle 3,1\rangle} V_{p_1} V_{p_2} V_{p_3}\rangle$. The other four fusion kernels all correspond to four-point functions with a degenerate field $V_{\langle 2,1\rangle}$.

**Degenerate fusion kernel at generic central charge**

In the case of a four-point function of the type $\langle V_{\langle 2,1\rangle} V_{p_1} V_{p_2} V_{p_3}\rangle$, the fusion kernel is given by the well-known formula

$$F_{p_1+\frac{\eta_1}{2}\beta, p_3+\frac{\eta_3}{2}\beta}\begin{bmatrix} p_1 & p_2 \\ \langle 2,1\rangle & p_3 \end{bmatrix} = \frac{\Gamma(1+2\eta_1\beta p_1)\Gamma(-2\eta_3\beta p_3)}{\prod_{\pm}\Gamma(\frac{1}{2}+\eta_1\beta p_1 \pm \beta p_2 - \eta_3\beta p_3)} \ . \tag{5.37}$$

Using the Pentagon relation (5.36), we deduce the fusion kernel for a four-point function of the type $\langle V_{\langle 3,1\rangle} V_{p_1} V_{p_2} V_{p_3}\rangle$:

$$F_{\epsilon_1,\epsilon_3} \underset{\epsilon_1,\epsilon_3\neq 0}{=} \frac{\Gamma(1+2\beta\epsilon_1 p_1)\Gamma(1+\beta^2+2\beta\epsilon_1 p_1)\Gamma(-2\beta\epsilon_3 p_3)\Gamma(-\beta^2-2\beta\epsilon_3 p_3)}{\prod_{\pm,\pm}\Gamma(\frac{1}{2}\pm\frac{1}{2}\beta^2+\beta\epsilon_1 p_1 \pm \beta p_2 - \beta\epsilon_3 p_3)} \ , \tag{5.38}$$

$$F_{\epsilon_1,0} \underset{\epsilon_1\neq 0}{=} \frac{\Gamma(\beta^2)}{\Gamma(2\beta^2)} \frac{\Gamma(1+2\beta\epsilon_1 p_1)\Gamma(1+\beta^2+2\beta\epsilon_1 p_1)\prod_{\pm}\Gamma(\beta^2\pm 2\beta p_3)}{\prod_{\pm,\pm}\Gamma(\frac{1}{2}+\frac{1}{2}\beta^2+\beta\epsilon_1 p_1 \pm \beta p_2 \pm \beta p_3)} \ , \tag{5.39}$$

$$F_{0,\epsilon_3} \underset{\epsilon_3\neq 0}{=} \frac{\Gamma(1-\beta^2)}{\Gamma(1-2\beta^2)} \frac{\Gamma(-2\beta\epsilon_3 p_3)\Gamma(-\beta^2-2\beta\epsilon_3 p_3)\prod_{\pm}\Gamma(1-\beta^2\pm 2\beta p_1)}{\prod_{\pm,\pm}\Gamma(\frac{1}{2}-\frac{1}{2}\beta^2\pm\beta p_1 \pm \beta p_2 - \beta\epsilon_3 p_3)} \ , \tag{5.40}$$

$$F_{0,0} = \frac{1}{\pi^2}\prod_{\pm}\Gamma(1-\beta^2\pm 2\beta p_1)\prod_{\pm}\Gamma(\beta^2\pm 2\beta p_3)$$

$$\times \left[ \cos(\pi\beta^2)\cos(\pi 2\beta p_2) + \cos(\pi 2\beta p_1)\cos(\pi 2\beta p_3) \right] \ , \tag{5.41}$$

where we introduced the notation $F_{\epsilon_1,\epsilon_3} = F_{p_1+\epsilon_1\beta, p_3+\epsilon_3\beta}\begin{bmatrix} p_1 & p_2 \\ \langle 3,1\rangle & p_3 \end{bmatrix}$.

**$c \to 1$ limit**

For generic momentums $p_i$, it is straightforward to compute the limit of the fusion kernel as $c \to 1$ i.e. $\beta \to 1$. There is only a cancellation of poles in $F_{0,\epsilon_3}$, which appears in $\lim_{\beta\to 1}\frac{\Gamma(1-\beta^2)}{\Gamma(1-2\beta^2)} = -2$.

Complications appear in the presence of degenerate fields, whose momentums become integer as $c \to 1$. The limit of the fusion kernel can then depend on how the momentums approach their $c = 1$ values. A degenerate field with momentum $k$ at $c = 1$ has a null vector at level $2k + 1$, therefore its continuation to $c \neq 1$ should be $V_{\langle 2k+1,1\rangle}$ or $V_{\langle 1,2k+1\rangle}$. We pick the first possibility, and find that the momentum behaves as

$$\beta p_{(2k+1,1)} = k + (\beta^2 - 1)(k + \tfrac{1}{2}) + O\left((\beta^2 - 1)^2\right) \ . \tag{5.42}$$

When it comes to a twist field of momentum $r \in \frac{1}{4} + \frac{1}{2}\mathbb{Z}$, the continuation that is compatible with the degenerate fields' fusion rules is $p_{(2r,0)}$, which behaves as

$$\beta p_{(2r,0)} = r + (\beta^2 - 1)r + O\left((\beta^2 - 1)^2\right) \ . \tag{5.43}$$

For vertex operators, we simply assume that the momentum is $\beta$-independent.

**Degenerate fusion kernels at $c = 1$**

The crossing symmetry equations for the chiral four-point functions $\langle I_1 I_{k_1} I_{k_2} I_{k_3} \rangle$ involve the fusion kernels $F_{\epsilon_1, \epsilon_3} \begin{bmatrix} k_1 & k_2 \\ 1 & k_3 \end{bmatrix}$ with $k_\alpha \in \mathbb{N}$. We compute these kernels as limits of the generic $c$ expressions (5.38) and (5.39), given the behaviour (5.42) of the degenerate momenta. We write the result under the assumptions $k_\alpha \in \mathbb{N}^*$ and $k_\alpha + k_\beta - k_\gamma \geq 0$. Cancellations between poles of Gamma functions lead to the finite expressions

$$
\begin{bmatrix} F_{+,+} & F_{-,+} \\ F_{+,0} & F_{-,0} \\ F_{+,-} & F_{-,-} \end{bmatrix} = \begin{bmatrix} \frac{\Gamma^{(2)}(1+2k_1)\Gamma^{(2)}(1+k_{23}^1)}{\Gamma^{(2)}(2+2k_3)\Gamma^{(2)}(k_{12}^3)} & \frac{\Gamma^{(2)}(1+k_{13}^2)\Gamma(k_{123})\Gamma(k_{123}+3)}{\Gamma^{(2)}(2k_3+2)\Gamma(-1+2k_1)\Gamma(2+2k_1)} \\ (-)^{k_{13}^2} \frac{\Gamma^{(2)}(1+2k_1)\Gamma(1+k_{23}^1)}{\Gamma(1+k_{123})\Gamma(1+k_{12}^3)\Gamma(1+k_{13}^2)} & -(-)^{k_{13}^2} \frac{\Gamma(1+k_{12}^3)\Gamma(1+k_{13}^2)\Gamma(k_{123})(1+k_{123})}{\Gamma(1+k_{23}^1)\Gamma(2k_1)\Gamma(2+2k_1)} \\ \frac{\Gamma^{(2)}(1+2k_1)\Gamma^{(2)}(-1+2k_3)}{\Gamma^{(2)}(k_{123})\Gamma^{(2)}(k_{13}^2)} & \frac{\Gamma^{(2)}(-1+2k_3)\Gamma^{(2)}(1+k_{12}^3)}{\Gamma(-1+2k_1)\Gamma(2+2k_1)\Gamma^{(2)}(k_{23}^1)} \end{bmatrix} ,
$$
(5.44)

where we introduce the notation

$$
\Gamma^{(2)}(x) = \Gamma(x)\Gamma(x+1) .
\tag{5.45}
$$

The crossing symmetry equations for the chiral four-point function $\langle I_1 V_p V_p I_k \rangle$ involve the fusion kernels $F_{0,\epsilon} \begin{bmatrix} p & p \\ 1 & k \end{bmatrix}$ with $k \in \mathbb{N}^*$ and $p \in \mathbb{C}$. We compute these fusion kernels as $c \to 1$ limits of generic $c$ expressions, and find

$$
\begin{bmatrix} F_{0,+} \\ F_{0,0} \\ F_{0,-} \end{bmatrix} = \begin{bmatrix} \frac{\Gamma^{(2)}(k+1)}{\Gamma(2k+2)^2} \prod_\pm \frac{\Gamma(\pm 2p)}{\Gamma(-k\pm 2p)} \\ 0 \\ -2 \frac{\Gamma^{(2)}(2k-1)}{\Gamma(k)^2} \prod_\pm \frac{\Gamma(\pm 2p)}{\Gamma(k\pm 2p)} \end{bmatrix} .
\tag{5.46}
$$

The crossing symmetry equations for the four-point function $\langle I_1 I_k T_{r_1} T_{r_2} \rangle$ involves the fusion kernels $F_{\epsilon_1, \epsilon} \begin{bmatrix} k & r_1 \\ 1 & r_2 \end{bmatrix}$ with $k \in \mathbb{N}^*$ and $r_1, r_2 \in \frac{1}{4} + \frac{1}{2}\mathbb{Z}$. We compute these fusion kernels as $c \to 1$ limits of generic $c$ expressions, and find

$$
\begin{bmatrix} F_{+,\epsilon} & F_{-,\epsilon} \\ F_{+,0} & F_{-,0} \end{bmatrix} = \begin{bmatrix} \frac{\Gamma^{(2)}(1+2k)\Gamma^{(2)}(-1-2\epsilon r_2)}{\prod_\pm \Gamma^{(2)}(k\pm r_1 - \epsilon r_2)} & \frac{\Gamma^{(2)}(1+k+\epsilon(r_2\pm_{\mathbb{Z}} r_1))\Gamma^{(2)}(-1-2\epsilon r_2)}{\Gamma(-1+2k)\Gamma(2+2k)\Gamma^{(2)}(-k-\epsilon(r_2 \mp_{\mathbb{Z}} r_1))} \\ \frac{\Gamma^{(2)}(1+2k)\prod_\pm \Gamma(1\pm 2r_2)}{\prod_{\pm,\pm}\Gamma(1+k\pm r_1 \pm r_2)} & -\frac{\prod_\pm \Gamma(1\pm 2r_2)\prod_\pm\Gamma(1+k\pm(r_1\pm_{\mathbb{Z}} r_2))}{\Gamma(-1+2k)\Gamma(2+2k)\prod_\pm\Gamma(1-k\pm(r_1\mp_{\mathbb{Z}} r_2))} \end{bmatrix} , \tag{5.47}
$$

where we assume $\epsilon \in \{+, -\}$, and we use the notation $r_1 \mp_{\mathbb{Z}} r_2$ defined in Eq. (4.17).

# 6 Conclusion and outlook

Solving the compactified free boson and the Ashkin–Teller model from the point of view of their affine symmetry algebra was not very difficult: building on earlier work, we only had to determine the signs of structure constants, and to check crossing symmetry of four-point functions of the type $\langle VVTT \rangle$.

We have then focused on finer observables in the same model, namely correlation functions of Virasoro primary fields. To compute them, we have developed a new flavour of the conformal bootstrap method, and written equations that relate the sought-after chiral structure constants to Virasoro fusion kernels. The resulting structure constants are written using the same Barnes' $G$-function that appears in Liouville theory at $c = 1$,

and also in the Virasoro fusion kernel at $c = 1$. However, the latter objects are not easily related to our structure constants, because they tend to be singular when momentums take degenerate values.

From the point of view of Virasoro symmetry, we have completely solved the compactified free boson and Ashkin–Teller model on the sphere. This means determining fusion rules and structure constants of all Virasoro primary fields, and checking crossing symmetry of their four-point functions. These solutions are based on a number of technical results on conformal blocks and fusion kernels at $c = 1$ and beyond, which may be of more general interest.

As an outlook, let us now sketch three perspectives that are opened by our results or techniques.

### Towards a solution of the 4-state Potts model

Although the Ashkin–Teller model with radius $R = 2$ is supposed to coincide with the 4-state Potts model, we are not claiming that we have solved the latter model. To do this, we would still need to organize the fields in representations of the symmetric group $S_4$, and to show that correlation functions are covariant under that symmetry.

At $R = 2$, the vertex sector momentums $p_{(n,w)} = n + \frac{w}{4}$ become degenerate if $w$ is even, and coincide with twist sector momentums if $w$ is odd. This allows the $S_4$ symmetry to mix the vertex sector with the degenerate and twist sectors. This also makes it clear that the $S_4$ symmetry does not commute with the affine symmetry. Solving the Ashkin–Teller model from the point of view of its Virasoro symmetry was therefore indeed necessary, but it is not yet sufficient. The mixing of the sectors must imply nontrivial identities between structure constants, and also between conformal blocks. Understanding these identities, and other properties of the model that emerge at $R = 2$, is left for future work.

### Degenerate conformal blocks at rational central charge

In order to compute the $c = 1$ conformal blocks and fusion kernels that we needed, we had to face the wider issue of understanding degenerate conformal blocks at rational values of the central charge. Our understanding is summarized in Conjecture 2, which passes a number of tests at $c = 1$. For other rational central charges, and in particular for minimal models, the conjecture's assumptions are probably too restrictive, and we should find a way to relax them.

This would be well worth the effort, as there is no known efficient way of computing conformal blocks in minimal models. These conformal blocks do obey BPZ differential equations, but the order of the equation depends on the particular block under consideration. It is not possible to write BPZ equations in general, and much less to solve them. In contrast, Zamolodchikov's recursive representation of conformal blocks is valid in generic cases, and can only simplify in special cases. However, the recursive representation becomes singular at rational central charges.

Using Conjecture 2, we have found examples of recursive representations for conformal blocks at $c = 1$. This raises the hope that recursive representations can be found at other rational central charges. This improves on the more pessimistic assessments of previous works [20, 21], which checked that the recursion had a finite limit as the central charge becomes rational, but did not find hints that the limit could be written explicitly. We have even managed to solve the recursion and find closed form expressions of the blocks as generalized theta functions, although this may be specific to our examples.

**Generalization to generic central charge**

It would be interesting to continue the Ashkin–Teller model to generic central charge, and obtain a simple, solvable, non-diagonal CFT. Of course, the $R = 2$ model has a known continuation, namely the $Q$-state Potts model, but with its complicated spectrum and large multiplicities, it does not count as a simple CFT.

Instead, we could start with the compactified free boson: this CFT can be continued to generic central charge, although the radius becomes quantized [9]. The next step would be to take its $\mathbb{Z}_2$ orbifold. If it was possible, this construction could even preserve the affine symmetry.

An important piece of the puzzle may be the continuation of the dimension $\frac{1}{16}$ of twist fields. In the $Q$-state Potts model, the continuation is $\Delta_{(0,\frac{1}{2})} = \frac{8-4\beta^2-3\beta^{-2}}{16}$, and this is also natural from the point of view of conformal blocks, as we saw in Section 5.2. On the other hand, from their geometric definition, $\mathbb{Z}_2$ twist fields at generic central charge have the dimension $\frac{c}{16} = \frac{13-6\beta^2-6\beta^{-2}}{16}$. Of course, it is not excluded that the model has several different continuations.

# Acknowledgements

We are grateful to Nina Javerzat for helpful comments and suggestions on the draft of this article. We thank the anonymous SciPost reviewers for their helpful suggestions, which can be viewed online.

This paper is partly a result of the ERC-SyG project, Recursive and Exact New Quantum Theory (ReNewQuantum) which received funding from the European Research Council (ERC) under the European Union's Horizon 2020 research and innovation programme under grant agreement No 810573. The work of NN is partly supported by Leading Research Center on Quantum Computing (Agreement no. 014/20), DFG projects CRC/TRR 191, and SFB/TRR 183.

# A    Conformal blocks and four-point functions

In this appendix we recall how affine symmetry allows us to find conformal blocks of vertex operators. Then we give a more detailed account of computing conformal blocks that involve twist fields. In Section A.2 we will then study a few simple four-point functions, with the aim of checking some basic features of the fusion rules.

## A.1    Derivation of some conformal blocks

In order to compute conformal blocks, we will study chiral correlation functions. The fields that appear in such correlation functions are labelled by their left-moving momentums.

**Vertex operators**

Affine Ward identities for a correlation function of the vertex operators follow from the OPE (3.3),

$$\left\langle J(z) \prod_i V_{p_i}(z_i) \right\rangle = \sum_j \frac{p_j}{z - z_j} \left\langle \prod_i V_{p_i}(z_i) \right\rangle . \tag{A.1}$$

Requirement that $J(z) \underset{z\to\infty}{=} O(z^{-2})$ leads to the neutrality condition $\sum_i p_i = 0$. Combining the Ward identities with relation

$$\partial V_p(w) = L_{-1}V_p(w) = 2J_0J_{-1}V_p(w) \;, \tag{A.2}$$

gives the Knizhnik–Zamolodchikov equation

$$\partial_j \left\langle \prod_i V_{p_i}(z_i) \right\rangle = \sum_{i\neq j} \frac{p_ip_j}{z_{ij}} \left\langle \prod_i V_{p_i}(z_i) \right\rangle \;. \tag{A.3}$$

The affine conformal block is then the unique solution that satisfies the normalization condition (2.13),

$$\widehat{\mathcal{F}}(z_i) = \prod_{i<j} z_{ij}^{2p_ip_j} \;. \tag{A.4}$$

The uniqueness of the solution corresponds to the fact that each intermediate channel contains a single operator, according to fusion rules $V_{p_1}\widehat{\times}V_{p_2} = V_{p_1+p_2}$.

**Two vertex operators and two twist fields**

From the OPE (4.2) of the current with the affine primary twist field, we deduce

$$\left\langle J(z)V_{p_1}(z_1)V_{p_2}(z_2)T_{\frac14}(z_3)T_{\frac14}(z_4) \right\rangle =$$
$$\frac{\sqrt{z_{13}z_{14}}\frac{p_1}{z-z_1} + \sqrt{z_{23}z_{24}}\frac{p_2}{z-z_2}}{\sqrt{(z-z_3)(z-z_4)}} \left\langle V_{p_1}(z_1)V_{p_2}(z_2)T_{\frac14}(z_3)T_{\frac14}(z_4) \right\rangle \;. \tag{A.5}$$

This chiral correlation function is completely determined by the analytic properties of $J(z)$. In particular, the denominator accounts for the branch cut stretching between the twist fields. The numerator has poles at the positions of vertex operators.

With the particular positions (2.14), our chiral correlation function becomes

$$\left\langle J(z)V_{p_1}(x)V_{p_2}(0)T_{\frac14}(\infty)T_{\frac14}(1) \right\rangle = \frac{\frac{p_2}{z} + \frac{p_1\sqrt{1-x}}{z-x}}{\sqrt{1-z}} \left\langle V_{p_1}(x)V_{p_2}(0)T_{\frac14}(\infty)T_{\frac14}(1) \right\rangle \;. \tag{A.6}$$

From this relation we compute how $J_{-1}$ acts on a vertex operator,

$$\left\langle (J_{-1}V_{p_1}(x))\, V_{p_2}(0)T_{\frac14}(\infty)T_{\frac14}(1) \right\rangle =$$
$$\oint_{z=x} \frac{dz}{2\pi i(z-x)} \left\langle J(z)V_{p_1}(x)V_{p_2}(0)T_{\frac14}(\infty)T_{\frac14}(1) \right\rangle =$$
$$\left( \frac{p_2}{x\sqrt{1-x}} + \frac{p_1}{2(1-x)} \right) \left\langle V_{p_1}(x)V_{p_2}(0)T_{\frac14}(\infty)T_{\frac14}(1) \right\rangle \;. \tag{A.7}$$

Combining this relation with Eq. (A.2) yields the first-order differential equation

$$\partial_x \log \left\langle V_{p_1}(x)V_{p_2}(1)T_{\frac14}(\infty)T_{\frac14}(1) \right\rangle = \frac{2p_1p_2}{x\sqrt{1-x}} + \frac{p_1^2}{1-x} \;. \tag{A.8}$$

The solution that respects the normalization condition (2.15) is

$$\widehat{\mathcal{F}}^{(s)}_{V_{1+2}|V_1V_2T_{\frac14}T_{\frac14}} = 16^{p_1p_2}(1-x)^{-p_1^2} \left( \frac{1-\sqrt{1-x}}{1+\sqrt{1-x}} \right)^{2p_1p_2} \;. \tag{A.9}$$

Since the $t$-channel conformal block is a solution of the same first-order differential equation, it is given by the same function albeit with a different normalization,

$$\widehat{\mathcal{F}}^{(t)}_{T_{\frac14}|V_1V_2T_{\frac14}T_{\frac14}} = (1-x)^{-p_1^2} \left( \frac{1-\sqrt{1-x}}{1+\sqrt{1-x}} \right)^{2p_1p_2} \;. \tag{A.10}$$

**Two vertex operators and two higher twist fields**

From the expression (4.5) of the higher twist field $T_{\frac{3}{4}}$ as an affine descendent of the basic twist field $T_{\frac{1}{4}}$, we deduce

$$\mathcal{F}^{(s)}_{V_{1+2}|V_1 V_2 T_{\frac{1}{4}} T_{\frac{3}{4}}} \propto 2 \oint_{z=1} \frac{dz}{2\pi i \sqrt{z-1}} \left\langle J(z) V_{p_1}(x) V_{p_2} T_{\frac{1}{4}}(\infty) T_{\frac{1}{4}}(1) \right\rangle . \tag{A.11}$$

Performing the integral and normalizing the result appropriately gives

$$\mathcal{F}^{(s)}_{V_{1+2}|V_1 V_2 T_{\frac{1}{4}} T_{\frac{3}{4}}} = \frac{p_2 + \frac{p_1}{\sqrt{1-x}}}{p_1 + p_2} \mathcal{F}^{(s)}_{V_{1+2}|V_1 V_2 T_{\frac{1}{4}} T_{\frac{1}{4}}} . \tag{A.12}$$

Conformal block with a second excited twist field $T_{\frac{5}{4}}$ can in principle be computed from its expression (4.6) as an affine descendent of $T_{\frac{1}{4}}$. However, it is technically simpler to use relation $T_{\frac{5}{4}} = 2J_{-\frac{3}{2}} T_{\frac{1}{4}} - \frac{4}{3} L_{-1} T_{\frac{3}{4}}$, which follows from that expression, together with the relation $L_{-1} T_{\frac{3}{4}} = J_{-\frac{3}{2}} T_{\frac{1}{4}} + 2J^3_{-1/2} T_{\frac{1}{4}}$. Omitting details of the computation we present the result

$$\mathcal{F}^{(s)}_{V_{1+2}|V_1 V_2 T_{\frac{1}{4}} T_{\frac{5}{4}}} =$$

$$\frac{1}{(p_1+p_2)(4(p_1+p_2)^2-1)} \left( p_2(4p_2^2-1) + \frac{12p_2^2 p_1}{\sqrt{1-x}} + \frac{12p_2 p_1^2}{1-x} + \frac{p_1(4p_1^2-1)}{(1-x)^{\frac{3}{2}}} \right)$$

$$\times \mathcal{F}^{(s)}_{V_{1+2}|V_1 V_2 T_{\frac{1}{4}} T_{\frac{1}{4}}} . \tag{A.13}$$

**Four twist fields**

We now turn to the correlation functions of four twist fields. Our presentation follows [5]. Unlike the vertex operators the twist fields are not eigenstates of $J_0$. Instead they are interrelated by the affine action. In particular from definition (4.5) one finds the following OPEs

$$J(z)T_{\frac{1}{4}}(w) = \frac{T_{\frac{3}{4}}(w)}{2\sqrt{z-w}} + O(\sqrt{z-w}) ,$$

$$J(z)T_{\frac{3}{4}}(w) = \frac{T_{\frac{1}{4}}(w)}{2(z-w)^{\frac{3}{2}}} + \frac{2\,\partial T_{\frac{1}{4}}(w)}{\sqrt{z-w}} + O(\sqrt{z-w}) , \tag{A.14}$$

where we have used $J_{\frac{1}{2}} T_{\frac{3}{4}} = \frac{1}{2} T_{\frac{1}{4}}$ and relation $J_{-\frac{1}{2}} T_{\frac{3}{4}} = 2J^2_{-\frac{1}{2}} T_{\frac{1}{4}} = L_{-1} T_{\frac{1}{4}} = \partial T_{\frac{1}{4}}$. Let us introduce four correlators

$$\mathcal{J}_0(z,x) = \langle J(z) T_{\frac{1}{4}} T_{\frac{1}{4}} T_{\frac{1}{4}} T_{\frac{1}{4}} \rangle , \qquad \mathcal{F}_0(x) = \langle T_{\frac{1}{4}} T_{\frac{1}{4}} T_{\frac{1}{4}} T_{\frac{1}{4}} \rangle ,$$

$$\mathcal{J}_1(z,x) = \langle J(z) T_{\frac{3}{4}} T_{\frac{1}{4}} T_{\frac{1}{4}} T_{\frac{1}{4}} \rangle , \qquad \mathcal{F}_1(x) = \langle T_{\frac{3}{4}} T_{\frac{1}{4}} T_{\frac{1}{4}} T_{\frac{1}{4}} \rangle . \tag{A.15}$$

Labels 0 and 1 refer to the correlation functions with the lowest and the first excited twist field at position $x$, respectively. Analytic form of $\mathcal{J}_0$ and $\mathcal{J}_1$ as functions of $z$ is fixed by OPEs (A.14)

$$\mathcal{J}_0(z,x) = \frac{A_0(x)}{\sqrt{z(z-x)(z-1)}}, \qquad \mathcal{J}_1(z,x) = \frac{\frac{A_1(x)}{z-x} + B_1(x)}{\sqrt{z(z-x)(z-1)}} , \tag{A.16}$$

with

$$A_0(x) = \frac{\sqrt{x(x-1)}}{2}\mathcal{F}_1(x), \qquad A_1(x) = \frac{\sqrt{x(x-1)}}{2}\mathcal{F}_0(x),$$

$$B_1(x) = 2\sqrt{x(x-1)}\left[\partial\mathcal{F}_0(x) + \frac{1}{8}\left(\frac{1}{x} + \frac{1}{x-1}\right)\mathcal{F}_0(x)\right] . \tag{A.17}$$

We have two equations (A.16) relating four correlation functions (A.15) which therefore do not fix the correlators unambiguously. This is due to the fact that fusion of two twist fields contains a spectrum of vertex operators $T_{\frac{1}{4}} \times T_{\frac{1}{4}} = \sum_p V_p$ and hence one has many conformal blocks satisfying these relations. To single out a conformal block with intermediate momentum $p$ we additionally impose

$$\oint_{0,x} dz\, \mathcal{J}_0(z,x) = 2\pi ip\, \mathcal{F}_0(x), \qquad \oint_{0,x} dz\, \mathcal{J}_1(z,x) = 2\pi ip\, \mathcal{F}_1(x) . \tag{A.18}$$

Now there are just enough equations to determine the solutions. Evaluating (A.18) explicitly one obtains

$$4A_0(x)K(x) = 2\pi ip\, \mathcal{F}_0(x), \qquad 4B_1(x)K(x) + 8A_1(x)\frac{K(x)}{dx} = 2\pi ip\, \mathcal{F}_1(x), \tag{A.19}$$

where

$$K(x) = \frac{1}{2}\int_0^1 \frac{dt}{\sqrt{t(1-t)(1-xt)}} \tag{A.20}$$

is the complete elliptic integral of the first kind. After a straightforward algebra one obtains equation for $\mathcal{F}_0(x)$

$$\frac{\partial}{\partial x}\log\left(\mathcal{F}_0(x)x^{1/8}(1-x)^{1/8}K^{1/2}(x)\right) = \frac{\pi^2 p^2}{4x(1-x)K^2(x)} . \tag{A.21}$$

Using identity

$$K(1-x)\frac{d}{dx}K(x) - K(x)\frac{d}{dx}K(1-x) = \frac{\pi}{4x(1-x)} , \tag{A.22}$$

one can verify that

$$\mathcal{F}_0(x) = \frac{16^{p^2}e^{-\pi p^2\frac{K(1-x)}{K(x)}}}{x^{1/8}(1-x)^{1/8}\sqrt{2K(x)/\pi}} \tag{A.23}$$

is a solution to (A.21) with correct normalization (2.13). It is convenient to introduce elliptic parameters

$$\tau = i\frac{K(1-x)}{K(x)}, \qquad q = e^{i\pi\tau} \tag{A.24}$$

and associated theta-constants defined by

$$\vartheta_1'(\tau) = \pi\sum_{n\in\mathbb{Z}}(-)^n(2n+1)q^{(n+\frac{1}{2})^2}, \qquad \vartheta_2(\tau) = \sum_{n\in\mathbb{Z}}q^{(n+\frac{1}{2})^2},$$

$$\vartheta_3(\tau) = \sum_{n\in\mathbb{Z}}q^{n^2}, \qquad \vartheta_4(\tau) = \sum_{n\in\mathbb{Z}}(-)^n q^{n^2} . \tag{A.25}$$

Using relation $K(x) = \frac{\pi}{2}\vartheta_3^2(q)$ conformal block (A.23) can be rewritten as

$$\widehat{\mathcal{F}}^{(s)}_{V_p|T_{\frac{1}{4}}T_{\frac{1}{4}}T_{\frac{1}{4}}T_{\frac{1}{4}}} = \frac{(16q)^{p^2}}{x^{1/8}(1-x)^{1/8}\vartheta_3(\tau)} \ . \tag{A.26}$$

The function $F_0(q)$ used in (4.25) is then

$$F_0(q) = \frac{1}{x^{1/8}(1-x)^{1/8}\vartheta_3(\tau)} \ . \tag{A.27}$$

Note that here we have restored the full notation for affine conformal blocks. Having found $\mathcal{F}_0(x)$ it is straightforward to also compute $\mathcal{F}_1(x)$

$$\widehat{\mathcal{F}}^{(s)}_{V_p|T_{\frac{3}{4}}T_{\frac{1}{4}}T_{\frac{1}{4}}T_{\frac{1}{4}}} = \frac{(16q)^{p^2}}{x^{5/8}(1-x)^{5/8}\vartheta_3(\tau)^3} \ . \tag{A.28}$$

Conformal blocks with other combinations of excited twist fields can be obtained in a similar way or using a more efficient technique [5]. Below we present several examples used in the main text without a derivation

$$\widehat{\mathcal{F}}^{(s)}_{V_p|T_{\frac{5}{4}}T_{\frac{1}{4}}T_{\frac{1}{4}}T_{\frac{1}{4}}} = \frac{(16q)^{p^2}}{x^{13/8}(1-x)^{13/8}\vartheta_3(\tau)^7}\frac{p^2 - q\frac{\partial}{\partial q}\log\vartheta_1'(\tau)}{p^2 - \frac{1}{4}}, \tag{A.29}$$

$$\widehat{\mathcal{F}}^{(s)}_{V_p|T_{\frac{3}{4}}T_{\frac{3}{4}}T_{\frac{1}{4}}T_{\frac{1}{4}}} = \frac{(16q)^{p^2}}{x^{9/8}(1-x)^{5/8}\vartheta_3(\tau)^5}\frac{p^2 - q\frac{\partial}{\partial q}\log\vartheta_2(\tau)}{p^2 - \frac{1}{4}}, \tag{A.30}$$

$$\widehat{\mathcal{F}}^{(s)}_{V_p|T_{\frac{3}{4}}T_{\frac{1}{4}}T_{\frac{1}{4}}T_{\frac{3}{4}}} = \frac{(16q)^{p^2}}{x^{5/8}(1-x)^{9/8}\vartheta_3(\tau)^5}\frac{p^2 - q\frac{\partial}{\partial q}\log\vartheta_4(\tau)}{p^2} \ . \tag{A.31}$$

Note that the ordering of the twist fields differs from [5] due to a difference in conventions for fixing their positions (2.14).

## A.2 Examples of four-point functions

In order to do some basic checks of fusion rules, let us consider a few examples of four-point functions, and work out their $s$-channel decompositions to the first few orders. In such low order calculations, and in our particular examples, the subtleties of Section 5 are not relevant, and we can compute degenerate conformal blocks from the $c = 1$ expression

$$\mathcal{F}^{(s)}_{\Delta_s}(\Delta_i|x) = x^{\Delta_s - \Delta_1 - \Delta_2}\left\{1 + c_1 x + c_2 x^2 + O(x^3)\right\} , \tag{A.32}$$

with the coefficients [9]

$$c_1 = \frac{(\Delta_s + \Delta_1 - \Delta_2)(\Delta_s + \Delta_4 - \Delta_3)}{2\Delta_s} , \tag{A.33}$$

$$c_2 = \frac{1}{(4\Delta_s - 1)^2}\begin{bmatrix}(\Delta_s + \Delta_1 - \Delta_2)_2 \\ \Delta_s + 2\Delta_1 - \Delta_2\end{bmatrix}^T\begin{bmatrix}2 + \frac{1}{4\Delta_s} & -3 \\ -3 & 4\Delta_s + 2\end{bmatrix}\begin{bmatrix}(\Delta_s + \Delta_4 - \Delta_3)_2 \\ \Delta_s + 2\Delta_4 - \Delta_3\end{bmatrix} , \tag{A.34}$$

where we use the notation $(\Delta)_2 = \Delta(\Delta + 1)$. To obtain $t$-channel blocks, we use the relation

$$\mathcal{F}^{(t)}_{\Delta_t}(\Delta_1, \Delta_2, \Delta_3, \Delta_4|x) = \mathcal{F}^{(s)}_{\Delta_t}(\Delta_1, \Delta_4, \Delta_3, \Delta_2|1 - x) \ . \tag{A.35}$$

(In this subsection, we write blocks as functions of conformal dimensions, not momentums.)

## Identity sector

Let us consider four-point functions of the first nontrivial Virasoro primary field in the identity sector, namely $I_{1,1} \propto J\bar{J}$. Given the self-OPE (3.1) of the current, we must have

$$\left\langle \prod_{i=1}^{4} J(z_i) \right\rangle \propto \frac{1}{z_{12}^2 z_{34}^2} + \frac{1}{z_{23}^2 z_{14}^2} + \frac{1}{z_{13}^2 z_{24}^2} \ , \tag{A.36}$$

or equivalently

$$\left\langle J(x)J(0)J(\infty)J(1) \right\rangle \propto \frac{1}{x^2} + \frac{1}{(1-x)^2} + 1 \underset{x \to 0}{=} \frac{1}{x^2}\left\{ 1 + 2x^2 + O(x^3) \right\} \ . \tag{A.37}$$

According to the fusion rule (3.17), we expect $s$-channel contributions from the two chiral fields $I_0$ and $I_2$. Since $I_2$ has dimension 4, it should not affect the leading three terms in the expansion near $x \to 0$. And indeed we find that these terms are accounted for by the conformal block

$$\mathcal{F}_0^{(s)}(1,1,1,1|x) = \frac{1}{x^2}\left\{ 1 + 2x^2 + O(x^3) \right\} \ . \tag{A.38}$$

While conformal blocks are generally singular at $\Delta_s = 0$, the block we are now considering is well-defined, provided we first set $\Delta_i = 1$ and then $\Delta_s = 0$.

## Mixed identity-twist four-point functions

Let us consider a four-point function of the type $\langle I_{1,1} I_{1,1} T^\epsilon T^\epsilon \rangle$. The left-moving factor of this four-point function is the chiral four-point function

$$\left\langle J(x)J(0)T^\epsilon(\infty)T^\epsilon(1) \right\rangle \propto \frac{1 - \frac{1}{2}x}{x^2\sqrt{1-x}} \underset{x \to 0}{=} \frac{1}{x^2}\left\{ 1 + \tfrac{1}{8}x^2 + O(x^3) \right\}$$

$$\underset{x \to 1}{=} \frac{1}{2\sqrt{1-x}}\left\{ 1 + 3(1-x) + 5(1-x)^2 + O((1-x)^3) \right\} \ . \tag{A.39}$$

In this four-point function, the OPEs determine the leading behaviour near the singularities $x = 0, 1, \infty$, which leaves the coefficient of the linear term undetermined in the numerator $1 - \frac{1}{2}x$. We determine this coefficient by requiring the vanishing of the $O(x)$ term in the expansion near $x = 0$. Then that expansion agrees with the conformal block

$$\mathcal{F}_0^{(s)}(1,1,\tfrac{1}{16},\tfrac{1}{16}|x) = \frac{1}{x^2}\left\{ 1 + \tfrac{1}{8}x^2 + O(x^3) \right\} \ . \tag{A.40}$$

Near $x = 1$, the fusion rule (4.15) predicts two terms, corresponding to the higher chiral twist fields $T^\epsilon_{\frac{3}{4}}, T^\epsilon_{\frac{5}{4}}$. The relevant conformal blocks are

$$\mathcal{F}_{\frac{9}{16}}^{(t)}(1,1,\tfrac{1}{16},\tfrac{1}{16}|x) = \frac{1}{\sqrt{1-x}}\left\{ 1 + 2(1-x) + 3(1-x)^2 + O((1-x)^3) \right\} \ , \tag{A.41}$$

$$\mathcal{F}_{\frac{25}{16}}^{(t)}(1,1,\tfrac{1}{16},\tfrac{1}{16}|x) = \frac{1}{\sqrt{1-x}}\left\{ (1-x) + 2(1-x)^2 + O((1-x)^3) \right\} \ . \tag{A.42}$$

There exists a linear combination of these blocks that agrees with our four-point function. This is a non-trivial check of the fusion rules.

The following chiral four-point function is determined (up to an overall constant factor) by its behaviour near the singularities $x = 0, 1, \infty$:

$$\left\langle J(x)J(0)T_{\frac{1}{4}}^{\epsilon}(\infty)T_{\frac{7}{4}}^{\epsilon}(1) \right\rangle \propto \frac{x^2}{(1-x)^{\frac{7}{2}}} \underset{x \to 0}{=} \frac{1}{x^2}\left\{ 1 + \tfrac{7}{2}x + \tfrac{63}{8}x^2 + O(x^3) \right\}$$

$$\underset{x \to 1}{=} \frac{1}{(1-x)^{\frac{7}{2}}}\left\{ 1 - 2(1-x) + (1-x)^2 + O((1-x)^3) \right\} . \quad \text{(A.43)}$$

According to the fusion rules, the expansion near $x = 0$ is the contribution of only one field $I_2$ of dimension 4, whose conformal block is

$$\mathcal{F}_4^{(s)}(1, 1, \tfrac{1}{16}, \tfrac{49}{16}|x) = \frac{1}{x^2}\left\{ 1 + \tfrac{7}{2}x + \tfrac{63}{8}x^2 + O(x^3) \right\} . \quad \text{(A.44)}$$

Likewise, the expansion near $x = 1$ only involves the field $T_{\frac{3}{4}}^{\epsilon}$, whose conformal block $\mathcal{F}_{\frac{9}{16}}^{(t)}(1, 1, \tfrac{1}{16}, \tfrac{49}{16}|x)$ agrees with our four-point function up to the order $O((1-x)^2)$.

**Mixed identity-vertex four-point functions**

Let us consider the chiral four-point function

$$\left\langle J(x)J(0)V_\Delta(\infty)V_\Delta(1) \right\rangle \propto \frac{1 - x + 2\Delta x^2}{x^2(1-x)} , \quad \text{(A.45)}$$

where primary fields are labelled by their conformal dimensions. The behaviour at $x = 0, 1, \infty$ determines this four-point function up to a polynomial of degree two, which we fix by requiring that the expansion at $x = 0$ agrees with the conformal block

$$\mathcal{F}_0^{(s)}(1, 1, \Delta, \Delta|x) = \frac{1}{x^2}\left\{ 1 + 2\Delta x^2 + O(x^3) \right\} . \quad \text{(A.46)}$$

The next block that is predicted by the fusion rules corresponds to the field $I_2$ whose dimension is 4, which does not contribute at this order. The expansion near $x = 1$ is then found to agree with the block

$$\mathcal{F}_\Delta^{(t)}(1, 1, \Delta, \Delta|x) = \frac{1}{1-x}\left\{ 1 + \tfrac{1}{2\Delta}(1-x) + \tfrac{1}{\Delta}(1-x)^2 + O((1-x)^3) \right\} . \quad \text{(A.47)}$$

**Vertex four-point functions**

Let us consider the vertex four-point function (4.23) in the case $(z_i) = (x, 0, \infty, 1)$,

$$\left\langle V_{(n,w)}(x)V_{(n,w)}(0)V_{(n',w')}(\infty)V_{(n',w')}(1) \right\rangle \propto \left| x^{-2p^2} \right|^2 \sum_{\pm} \left| (1-x)^{\pm 2pp'} \right|^2 . \quad \text{(A.48)}$$

According to the fusion rule (4.11), only the identity sector contributes in the $s$-channel. Up to the order $O(x^2)$ in the expansion near $x = 0$, the only contributing fields are $I_{0,0}$ and $I_{1,1}$, and the relevant conformal blocks are

$$\mathcal{F}_0(\Delta, \Delta, \Delta', \Delta'|x) = x^{-2\Delta}\left\{ 1 + 2\Delta\Delta' x^2 + O(x^3) \right\} , \quad \text{(A.49)}$$

$$\mathcal{F}_1(\Delta, \Delta, \Delta', \Delta'|x) = x^{-2\Delta}\left\{ x + \tfrac{1}{2}x^2 + O(x^3) \right\} . \quad \text{(A.50)}$$

And our four-point function can indeed be written as a combination of such blocks, schematically

$$\left\langle VVV'V' \right\rangle \propto |\mathcal{F}_0(x)|^2 + 4pp'\bar{p}\bar{p}' |\mathcal{F}_1(x)|^2 + O(x^3) . \quad \text{(A.51)}$$

# B    Discrete fusion transformations of Virasoro blocks

## B.1    The Virasoro fusion kernel at $c = 1$

### Continuous fusion transformation

The $s$- and $t$-channel Virasoro conformal blocks are related by the fusion transformation

$$\mathcal{F}^{(s)}_{p_s|1234} = \int_{\mathcal{C}} dp_t \ F_{p_s p_t} \begin{bmatrix} 2 & 3 \\ 1 & 4 \end{bmatrix} \mathcal{F}^{(t)}_{p_t|1234} \ , \tag{B.1}$$

where the fusion kernel $F_{p_s p_t} \begin{bmatrix} 2 & 3 \\ 1 & 4 \end{bmatrix}$ only depends on the fields' dimensions, and not on their positions. The fusion kernel is much simpler at $c = 1$ than at generic $c$, and can be written as [22]

$$F_{p_s p_t} \begin{bmatrix} 2 & 3 \\ 1 & 4 \end{bmatrix} = \mu \frac{C(p_s|p_1, p_2, p_3, p_4)}{C(p_t| - p_3, p_2, -p_1, p_4)} \prod_{i=1}^{4} \frac{\widehat{G}(\omega_+ + \nu_k)}{\widehat{G}(\omega_+ + \lambda_k)}, \tag{B.2}$$

where $\widehat{G}(z) = \frac{G(1+z)}{G(1-z)}$, and we define

$$C(p_s|p_1, p_2, p_3, p_4) = \frac{\prod_{\epsilon = \pm} G(1 + 2\epsilon p_s)}{\prod_{\epsilon, \epsilon' = \pm} G(1 + \epsilon p_s + \epsilon' p_2 + \epsilon\epsilon' p_1) G(1 + \epsilon p_s + \epsilon' p_3 + \epsilon\epsilon' p_4)} \ . \tag{B.3}$$

The parameters $\nu_k$ and $\lambda_k$ in the arguments of $\widehat{G}$-functions are defined as

$$\nu_1 = p_s + p_1 + p_2, \qquad\qquad \lambda_1 = p_1 + p_2 + p_3 + p_4, \tag{B.4}$$
$$\nu_2 = p_s + p_3 + p_4, \qquad\qquad \lambda_2 = p_s + p_t + p_1 + p_3, \tag{B.5}$$
$$\nu_3 = p_t + p_1 + p_4, \qquad\qquad \lambda_3 = p_s + p_t + p_2 + p_4, \tag{B.6}$$
$$\nu_4 = p_t + p_2 + p_3, \qquad\qquad \lambda_4 = 0 \ , \tag{B.7}$$

and we also introduce

$$\sigma = \frac{1}{2} \sum_{i=1}^{4} \nu_i = \frac{1}{2} \sum_{i=1}^{4} \lambda_i \ . \tag{B.8}$$

To define $\omega_+$ and $\mu$ in Eq. (B.2), we introduce the notations

$$c_i = 2 \cos(2\pi p_i) \quad , \quad s_i = 2 \sin(2\pi p_i) \ , \tag{B.9}$$

and the combinations

$$\omega_{12} = c_1 c_2 + c_3 c_4, \qquad \omega_{23} = c_2 c_3 + c_1 c_4, \qquad \omega_{13} = c_1 c_3 + c_2 c_4, \tag{B.10}$$
$$\omega_4 = \prod_{i=1}^{4} c_i + \sum_{i=1}^{4} c_i^2. \tag{B.11}$$

Using these variables we define $q_{13}$ as a solution of the quadratic equation

$$\frac{1}{4} q_{13}^2 - \frac{1}{4}(c_s c_t - \omega_{13})^2 + c_s^2 + c_t^2 - \omega_{12} c_s - \omega_{23} c_t + \omega_4 - 4 = 0 \ . \tag{B.12}$$

Then the prefactor $\mu$ in (B.2) is

$$\mu = -\frac{s_s s_t}{q_{13}} \ , \tag{B.13}$$

while the quantity $\omega_+$ is defined by

$$e^{2\pi i \omega_+} = \frac{s_s s_t + s_2 s_4 + s_1 s_3 + q_{13}}{2 \sum_{i=1}^4 \left(e^{2\pi i(\sigma - \nu_k)} - e^{2\pi i(\sigma - \lambda_k)}\right)} \ . \tag{B.14}$$

There are three ambiguities in the definition (B.2) of the fusion kernel:

1. The choice of a solution of the quadratic equation (B.12), which will not affect the fusion transformation.

2. The integration contour $\mathcal{C}$ in (B.1) should be chosen as $\mathbb{R} + i\Lambda$ with $\Lambda$ sufficiently large so that all singularities of the integrand lie below the contour.

3. The quantity $\omega_+$ is only defined modulo an integer. However, the identity

$$\widehat{G}(z + 1) = -\frac{\pi}{\sin(\pi z)} \widehat{G}(z) \ , \tag{B.15}$$

and the explicit forms of $\nu_i, \lambda_i$ ensure that the fusion kernel is invariant under integer shifts of $\omega_+$.

**Discrete fusion transformation**

Let us assume that the integrand of the fusion transformation (B.1) is meromorphic in $p_t$, with simple poles on the real $p_t$-line. Since the integrand is odd in $p_t$, we have

$$\mathcal{F}^{(s)}_{p_s} = \frac{1}{2}\left(\int_{\mathbb{R}+i\Lambda} - \int_{\mathbb{R}-i\Lambda}\right) dp_t \ F_{p_s p_t} \mathcal{F}^{(t)}_{p_t} = -\pi i \sum_{k \in \text{Poles}} \operatorname*{Res}_{p_t = k} F_{p_s p_t} \mathcal{F}^{(t)}_{p_t} \ . \tag{B.16}$$

This is how the continuous fusion transformation can become discrete, in agreement with the discrete fusion rules of the free boson and Ashkin–Teller models.

## B.2 Discrete fusion of higher twist fields

In order to determine chiral structure constants of higher twist fields in Section 4.4, we need some particular fusion transformations.

**Simplification of the fusion kernel**

Consider the fusion transformation

$$\mathcal{F}^{(s)}_{p_1+p_2|p_1 p_2 r_2 r_1} = \int_{\mathcal{C}} F_{p_1+p_2, p_t} \begin{bmatrix} p_2 & r_2 \\ p_1 & r_1 \end{bmatrix} \mathcal{F}^{(t)}_{p_t|p_1 p_2 r_2 r_1} \ , \tag{B.17}$$

which involves the fusion kernel (B.2) with momentums

$$(p_3, p_4) = (r_2, r_1) \quad \text{with} \quad r_1, r_2 \in \frac{1}{4} + \frac{1}{2}\mathbb{N} \ , \qquad p_s = -p_1 - p_2 \ . \tag{B.18}$$

With the help of the trigonometric identity

$$x + y + z = 0 \quad \Rightarrow \quad 2\cos x \cos y \cos z + 1 = \cos^2 x + \cos^2 y + \cos^2 z \ , \tag{B.19}$$

the quadratic equation (B.12) simplifies, and we find

$$q_{13} = 4i\cos(2\pi p_t)\sin(2\pi(p_1 + p_2)) \quad , \quad \omega_+ = 0 \ , \tag{B.20}$$

which leads to

$$\mu = -i\tan(2\pi p_t) \ . \tag{B.21}$$

This has simple poles for $p_t = r \in \frac{1}{4} + \frac{1}{2}\mathbb{Z}$, with the residues

$$-2\pi i \operatorname*{Res}_{p_t=k} \mu = 1 \ . \tag{B.22}$$

Accepting for an instant that only these simple poles contribute, we obtain the discrete fusion transformation

$$\boxed{\mathcal{F}^{(s)}_{p_1+p_2} = \sum_{r\in\frac{1}{4}+\frac{1}{2}\mathbb{N}} F_{p_1+p_2,r}\mathcal{F}^{(t)}_r} \ . \tag{B.23}$$

We compute the fusion kernel with the help of the duplication formula (2.28),

$$F_{p_1+p_2,r} = 16^{p_1 p_2} \prod_{\pm} \frac{G(1\pm\frac{1}{2})}{G(1\pm 2r)}$$

$$\times \prod_{\pm,\pm} \frac{G(1+p_1+p_2\pm\frac{1}{4}\pm\frac{1}{4})}{G(1+p_1+p_2\pm r_1\pm r_2)} \prod_{i=1}^{2}\prod_{\pm,\pm} \frac{G(1+p_i\pm r\pm r_i)}{G(1+p_i\pm\frac{1}{4}\pm\frac{1}{4})} \ . \tag{B.24}$$

In the case $r_1 = r_2 = p_t = \frac{1}{4}$, we recover the affine fusion kernel element $F_{p_1+p_2,\frac{1}{4}} = 16^{p_1 p_2}$.

**Cancellation of spurious poles**

The fusion kernel has not only simple poles for $p_t \in \frac{1}{4} + \frac{1}{2}\mathbb{Z}$, but also higher order poles for $p_t \in \frac{1}{2}\mathbb{Z}$. This is due to a factor $\frac{1}{\prod_\pm G(1\pm 2p_t)}$, which appears as $\frac{1}{\prod_\pm G(1\pm 2r)}$ in Eq. (B.24). We may therefore worry that the sum of residues includes contributions with half-integer momentums, which correspond to degenerate fields. However, the fusion rules forbid such contributions. The idea is that $t$-channel conformal blocks also have poles for $p_t \in \frac{1}{2}\mathbb{Z}$, whose contributions have to cancel the contributions from the fusion kernel. Such cancellation of poles, due to an interplay between structure constants and conformal blocks, are known to occur in analytically solvable CFTs [21]. In our case, let us demonstrate the cancellation mechanism in a simple example. We recall the leading terms of the $t$-channel conformal block,

$$\mathcal{F}^{(t)}_{p_t|p_1 p_2 r_2 r_1} = (1-z)^{p_t^2-p_1^2-r_1^2}\left(1 + \frac{\prod_{i=1}^{2}(p_t^2+p_i^2-r_i^2)}{2p_t^2}(1-z) + \cdots\right) \ . \tag{B.25}$$

Due to the $O(1-z)$ term of the conformal block, the fusion transformation's integrand (B.17) has a simple pole at $p_t = 0$. Neglecting $p_t$-independent factors from the fusion kernel, the residue is

$$\operatorname*{Res}_{p_t=0} F_{p_1+p_2,p_t}\mathcal{F}^{(t)}_{p_t} \propto \frac{1}{2}(1-z)^{1-p_1^2-r_1^2}\prod_{i=1}^{2}(p_i^2-r_i^2)\prod_{\pm}G(1+p_i\pm r_i)^2 + \cdots \ . \tag{B.26}$$

This should be compared with the residue of the simple pole at $p_t = 1$. We neglect the same $p_t$-independent prefactors, and focus again on the leading contribution as $z \to 1$,

$$\operatorname*{Res}_{p_t=1} F_{p_1+p_2,p_t} \mathcal{F}_{p_t}^{(t)} \propto -\frac{1}{4}(1-z)^{1-p_1^2-r_1^2} \prod_{i=1}^{2}\prod_{\pm,\pm} G(1 + p_i \pm 1 \pm r_i) + \cdots . \tag{B.27}$$

It remains to use the identity $\prod_{\pm} G(x \pm 1) = (x-1)G(x)^2$, for seeing that the residue at $p_t = 0$ cancels with the sum of the residues at $p_t = -1, 1$.

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
