# Peer review of "Analytic conformal bootstrap and Virasoro primary fields in the Ashkin-Teller model"

_SciPost Physics_

## Round 1 · Referee Report · Anonymous · 2021-8-17

Strengths

1. Very thorough and detailed.
2. Contains a conjecture which might be of interest also to readers who are not necessarily interested in the specific model.

Weaknesses

1. Very very technical.
2. Presentation could be improved.

Report

In this papers the authors study the Ashkin-Teller model. This has been solved from the point of view of the affine algebra, meaning that the conformal data in terms of the affine primaries is known. Conformal data for the Virasoro primaries can then be obtained, in principle. However, this becomes rather technical; the authors study directly correlation functions of Virasoro primaries. By imposing crossing symmetry of four point functions, the three point function coefficients are found.
The paper also contains a pair of conjectures on how to use Zamolodchikov’s recursion relation for the Virasoro blocks in case of degenerate operators. Naively, this is ill defined. The authors instead propose a precise way of deforming the theory so that the correct Virasoro blocks can be obtained. This might be of interest also to readers who are not necessarily interested in the Ashkin-Teller model.
I find the paper very thorough and rather technical. Most of the edits I suggest concern notation and presentation of the paper. Exceptions are the more important points 5 and 6, which deal with the precise recipe of Conjecture 2.

Requested changes

1. I have some remarks about the notation of section 2: starting from (2.11), the summation variable $s$ should be renamed since $s$ appears also as the letter indicating that we are expanding in the $s$-channel. (2.19) could be made leaner by replacing $\Delta_j \to j$, as is done in most of the section. Should (2.20) include $\Delta_t,\bar \Delta_t \in \text{spectrum} $ or something of the kind? Does (2.26) need a $\pm$ in front?

2. I find the minus sign difference between (3.39) and (3.46) confusing. Since the sign is not determined, can we choose it to be the same in these two expressions?

3. A clearer explaination of the notation of (3.41) would help (I mean which operator is at which coordinate in $F_{0,\epsilon} \big[\ldots\big]$)

4. I imagine that (4.19) could also be zero for some values of the $(n_i,m_j)$, given equation (4.10). Is that right? If so, it should be indicated in (4.19).

5. The recipe of Conjecture 2 doesn't specify which degenerate representation we should consider. There are several $\mathcal{R}^{(c)}$ who have the same number of states as $\mathcal{R}$ at generic central charge. The question of which $\mathcal{R}^{(c)}$ one must choose should be addressed, but it isn't. For example, above eq (5.16), the authors choose one of the two possible structures, but it's not explained why.

6. Probably related to the previous point, in which cases did the authors check the validity of the conjectures? Which conformal blocks can be computed explicitly and then compared to the conjecture? Is the check up to some level? It would be good to include this.

7.It should be explained what the pentagon diagram of (5.32) symbolizes so that a reader not familiar with it could understanding without looking at reference [9].

---

## Round 1 · Referee Report · Anonymous · 2021-8-29

Strengths

1. Contains novel conjectures regarding the computation of degenerate conformal blocks that may find other applications.
2. Very thorough in its presentation of technical results.

Weaknesses

1. Rather technical subject matter.

Report

In this paper the authors study correlation functions of local operators in the Ashkin-Teller model, the $\mathbb{Z}_2$ orbifold of the $c=1$ free boson CFT.

The authors begin with a review of correlation functions in the unorbifolded free boson theory. This includes an extended discussion of correlation functions of degenerate Virasoro primary operators that are not affine primaries. Although the conformal data of the latter are completely determined by those of the current algebra primaries via the affine symmetry, extracting this conformal data in this way in practice is highly tedious. The authors circumvent this by imposing crossing symmetry of the correlators of these Virasoro primaries directly, although this still turns out to be a rather technical exercise due to analytic subtleties of the conformal blocks when the dimensions correspond to those of degenerate representations.

The authors then proceed to study correlation functions in the orbifolded theory. This includes a, to the best of my knowledge, novel determination of structure constants involved in mixed four-point functions of affine primaries in the untwisted sector and twist fields. Together with the known four-point functions of affine primaries in the untwisted sector and of twist fields, this establishes consistency of the theory on the sphere. The authors also study correlation functions of degenerate Virasoro primaries and current algebra descendants of the twist fields, encountering similar subtleties as in the degenerate correlators of the free boson.

In studying the correlators of degenerate Virasoro primaries, the authors must compute the corresponding degenerate conformal blocks (or fusion kernels). In practice, the computation of degenerate blocks is subtle. The reason for this is that methods for the computation of generic conformal blocks view the blocks as meromorphic functions of (say) the central charge, with poles appearing when internal states correspond to degenerate representations of the Virasoro algebra (signalling the presence of null states). The authors propose a concrete method to determine degenerate conformal blocks as limits of generic conformal blocks in the form of two conjectures. These proposals might be of broader interest than the results on crossing symmetry in the Ashkin-Teller model.

The paper concerns rather technical subject matter but is well-written and very thorough in the presentation of its results. I recommend the publication of this paper in SciPost.

Requested changes

1. The notation $k_{123}$ and $k^1_{23}$ are undefined when used in (3.49). Although they are defined after (5.42), these definitions should be moved up to their first use in the main text.

2. The authors should discuss the extent to which they have checked their conjectures regarding degenerate conformal blocks as limits of generic conformal blocks.

---

## Round 2 · Referee Report · Anonymous · 2021-9-8

Report

With the expanded discussion on the scope and validity of the authors' conjectures regarding the computation of degenerate conformal blocks as limits of generic conformal blocks, I am happy to recommend the publication of this paper in SciPost.

---

## Round 2 · Author Response

We have added clarification following the reviewers' suggestions, in particular in Section 5.1.

---

## Round 2 · List of Changes

Answer to comments by Reviewer 1:

1. We have added a clarification about the notations after (2.12). In (2.19) and (2.20) we use the notation $\Delta_j$ in order to emphasize that the Virasoro blocks and fusion kernels are universal quantities that only depend on conformal dimensions, as we explain after (2.19). Eq. (2.20) should not include any restiction on $(\Delta_t,\bar\Delta_t)$, as we now clarify after that equation. We have added $\pm$ in Eq. (2.26) as suggested, in order to display the sign ambiguity explicitly.

2. As suggested, we have flipped the sign of (3.35), and consequently also of (3.39), (3.40), (4.42). This has allowed us to simplify the discussion of the sign ambiguity after (3.46), and to state after (4.60) that we have agreement with (4.40)-(4.43), and not just agreement up to signs.

3. We have tried to clarify this issue by consistently using the matrix notation for the fusion kernel, starting with (2.19). This change of noation has also affected Eqs. (B.1), (B.2), (B.17).

4. Our point of view is that we should not rely on the structure constants to impose fusion rules. In the OPE (2.8) and decomposition (2.11), fusion rules are imposed at the level of the summation instead. However, fusion rules were missing from the three-point function (2.6): we have now added them using an explicit prefactor, and added an explanation after that equation. This allows structure constants such as (4.19) to remain simple.

5. We believe that this choice does not matter. We have made this clearer in Conjecture 2 by using the word 'any' in 'Let $\mathcal{R}^{(c)}$ be any degenerate representation at generic $c$ that has the same number of states...'. In the text before (5.17), we have also stated explicitly that the choice of one of the two possible representations does not matter.

7. We have added explanations after the pentagon diagram, including the new diagram (5.34).

Answer to comments by Reviewer 2:

1. Following the suggestion, we have moved up the definitions of these notations.

Answer to both reviewers:

Following the convergent requests of both reviewers, we have expanded the paragraph 'Scope and validity of the conjectures' in Section 5.1, in order to discuss the evidence in more detail. In particular, the new Eq. (5.15) lists the conformal blocks for which we have evidence. We do not mention direct pedestrian tests of Conjecture 2, as they are limited to low levels, and therefore quite weak.

Answer to Editor's request:

We have changed the Bibtex style to the SciPost style, which displays journal references.

You are currently on this page

Resubmission 2106.15132v2 on 2 September 2021

---

## Editorial Decision

publication_decision_taken:_accept